# XRPO: Pushing the limits of GRPO with Targeted Exploration and Exploitation

## Abstract

Reinforcement learning algorithms such as GRPO have driven recent advances in large language model (LLM) reasoning. While scaling the number of rollouts stabilizes training, existing approaches suffer from limited exploration on challenging prompts and leave informative feedback signals underexploited, due to context-independent rollout allocation across prompts (e.g., generating 16 rollouts per prompt) and relying heavily on sparse rewards. This paper presents *XRPO* (eXplore–eXploit GRPO), a unified framework that recasts policy optimization through the principled lens of rollout exploration–exploitation. To enhance exploration, *XRPO* introduces a mathematically grounded rollout allocator that adaptively prioritizes prompts with higher potential for uncertainty reduction. It further addresses stagnation on zero-reward prompts through an in-context seeding strategy that injects curated exemplars, steering the model into more difficult reasoning trajectories. To strengthen exploitation, *XRPO* develops a group-relative, novelty-aware advantage sharpening mechanism that leverages sequence likelihoods to amplify low-probability yet correct responses, thereby extending the policy's reach beyond sparse rewards. Experiments across diverse math and coding benchmarks on both reasoning and non-reasoning models demonstrate that *XRPO* outperforms existing advances (e.g., GRPO and GSPO) up to 4% pass@1 and 6% cons@32, while accelerating training convergence by up to $2.7\times$.

## 1 Introduction

Recent breakthroughs in applying reinforcement learning (RL) to large language models, such as GPT-o3 (OpenAI, 2025), Qwen3 (Yang et al., 2025a), and Deepseek-R1 (Guo et al., 2025), have demonstrated its effectiveness in enhancing reasoning capabilities. A key driver of this progress has been reinforcement learning with verifiable rewards (RLVR), where models receive rule-based numerical feedback on their generations. RLVR, exemplified by GRPO (Shao et al., 2024) and its very recent extensions (e.g., GSPO (Zheng et al., 2025a)), has emerged as a primary pathway for achieving these breakthroughs.

Despite rapid progress, RLVR continues to face persistent challenges—most notably slow training and sparse feedback—that form fundamental bottlenecks to both efficiency and quality. We categorize these challenges through the lens of exploration and exploitation: expanding exploration into uncertain rollout regions, and maximizing exploitation of known informative behaviors:

1. *Under-exploration of valuable rollouts in generation.* Existing methods (e.g., GRPO and GSPO) use static rollout allocation across prompts (e.g., generating 16 rollouts per prompt), which dilutes valuable signals from high-reward-variance rollouts and leaves zero-accuracy prompts underexplored, whose mastery is critical for surpassing current performance limits. Recent dynamic sampling approaches attempt to gather learning signals by over-sampling or discarding prompts with accuracies of 1 or 0 (Yu et al., 2025). However, despite large computational overhead (e.g., generating multiple rollouts and only retaining a few (Hou et al., 2025)), these methods lack differentiated exploration and ignore how prompts vary in their potential to expand the model's decision boundary. Moreover, discarding zero-accuracy prompts, often hard questions beyond the model's current capability, risks the model never pushing past its limits.

2. *Under-exploitation of trajectory signals in rewards.* Simple rule-based rewards (e.g., 1 for correct response, otherwise 0) collapse distinctions among rollouts. Yet, the rich information em-

bedded in generation trajectories remains largely underexploited, which suppresses the model's ability to explore the broader decision space effectively. Recent advances have attempted to enrich exploration via step-wise dense rewards and tree sampling (Hou et al., 2025; Yang et al., 2025b), but these approaches incur high overhead (e.g., sampling many additional rollouts) and still rely on coarse heuristics to model the sampling space.

To address these limitations, we propose *XRPO* (eXplore–eXploit GRPO), a novel rollout optimization strategy that systematically balances exploration and exploitation. By promoting informative rollout exploration while sharpening exploitation reward signals, *XRPO* effectively breaks through the edge of model capability, leading to stable and higher-quality RLHF training. Our contributions are as follows:

1. *Novel Hierarchical Rollout Exploration*: We introduce a hierarchical rollout planner that adaptively allocates rollouts based on uncertainty reduction and exploration bonuses. This allows the rollout policy to focus on high-variance prompts near the decision boundary where additional rollouts are most informative. To address degenerate groups where all responses fail and gradients vanish, we further seed these hard prompts with curated in-context examples drawn from an evolving corpus of rollouts with verified successes over training. This combination ensures that both ambiguous and unsolved prompts contribute non-trivial learning signals, breaking symmetry and expanding the effective training frontier.

2. *Novelty-Guided Advantage Sharpening*: Beyond rollout exploration, *XRPO* improves exploitation of successful rollouts by introducing a sequence-level novelty measure. Rollouts that are correct yet atypical under the model's own distribution receive an additional entropy-inspired bonus. This mechanism both distinguishes among superficially similar successes and promotes generalization to underexplored reasoning paths, counteracting the homogenization imposed by rule-based sparse rewards.

3. *Comprehensive Evaluation*: We conduct extensive experiments on math reasoning and code generation benchmarks. *XRPO* consistently outperforms vanilla GRPO and the very recent advances by up to 4% pass@1. *XRPO* also improves sample efficiency by more than a factor of two, and achieves higher task success rates under the same rollout budgets. These results validate the effectiveness of *XRPO* 's principled balance between exploration and exploitation.

## 2 RELATED WORKS

**Reinforcement Learning for LLMs.** Recent advances in RLHF (DeepSeek-AI et al., 2025; Team, 2025; Li et al., 2025a) introduce RLVR, leveraging verifiable numerical signals to enhance reasoning. RLVR typically relies on sparse, rule-based numerical rewards, requiring many rollouts per prompt to estimate trajectory advantages, which introduces substantial training overhead (Gandhi et al., 2025; Xi et al., 2024). Methods such as GSPO (Zheng et al., 2025b) refine reward importance ratios using sequence likelihood and perform sequence-level clipping, but still under-explore rollouts critical to the frontier of model capability. Our approach addresses these limitations by dynamically allocating rollout resources with a mathematically grounded allocator and evaluating responses based on relative novelty rather than correctness alone.

**Efficient Data Selection for LLMs.** The rise of RLVR has motivated specialized data selection techniques, especially for GRPO-based RL training (Fatemi et al., 2025; Li et al., 2025b; Wang et al., 2025b). For instance, DAPO (Yu et al., 2025) improves gradient efficiency via dynamic sampling, over-sampling informative prompts while filtering out fully correct or fully incorrect ones. GRESO (Zheng et al., 2025c) uses prior reward training dynamics to bypass uninformative prompts. In contrast, *XRPO* dynamically allocates rollout budgets for the given prompts and across prompts in a batch at runtime, generating more valuable trajectories for both exploration and learning.

**Enforced Self-Refinement Fine-Tuning.** Forcing LLMs to correct or refine their own outputs has been shown to improve accuracy and, in some cases, enhance self-reflection. One approach uses Natural Language Feedback (NLF) (Hancock et al., 2019; Wang et al., 2025c; Chen et al., 2024) from human annotators or stronger models, which can be costly. For example, Critique-GRPO (Zhang et al., 2025a) employs a stronger model (e.g., GPT-4o) to provide informative critiques to the actor model. Another approach prompts the model to iteratively review and correct its responses via

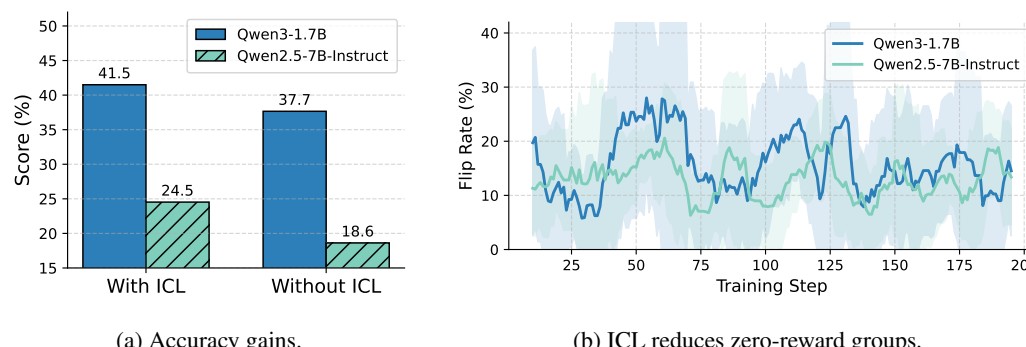

(a) Accuracy gains.                    (b) ICL reduces zero-reward groups.

Figure 1: Impact of ICL Seeding. **(a)** ICL significantly improves performance on the DAPO training dataset. **(b)** ICL recovers a significant portion of previously unsolved prompts, improving generalization to harder problems.

multi-turn RL (Chen et al., 2023; Kumar et al., 2024; Huang et al., 2023), offering explicit correctness feedback at each step. To refine performance on hard prompts with near-zero accuracy and scarce correct rollout histories, we adopt In-Context Learning (ICL) seeding with successful examples from related tasks, representing, to our knowledge, the first demonstration of ICL's effectiveness within an RL framework.

## 3 BACKGROUND AND MOTIVATION

### 3.1 GROUP RELATIVE POLICY OPTIMIZATION

Group Relative Policy Optimization (GRPO) (Shao et al., 2024) adapts Proximal Policy Optimization (PPO) (Schulman et al., 2017) with rule-based reward functions, with the following training objective:

$$\mathcal{J}^{\text{GRPO}}(\theta) = \mathbb{E}_{(q,\mathbf{o}) \sim \mathcal{D}_{\theta_{\text{old}}}} \left[ \frac{1}{G} \sum_{i=1}^{G} \min\left( \rho_i(\theta) A_i, \ \text{clip}\left( \rho_i(\theta), 1 - \epsilon, 1 + \epsilon \right) A_i \right) - \beta D_{\text{KL}}(\pi_\theta || \pi_{\text{ref}}) \right]$$

$$\text{where} \quad \rho_i(\theta) = \frac{\pi_\theta(o_i \mid q)}{\pi_{\theta_{\text{old}}}(o_i \mid q)}, \quad A_i = \frac{R(q, o_i) - \text{mean}(\{R(q, o_1), \ldots, R(q, o_G)\})}{\text{std}(\{R(q, o_1), \ldots, R(q, o_G)\})}.$$

Here, $\mathbf{o} = \{o_1, \ldots, o_G\}$ represents $G$ response rollouts sampled from $\pi_{\theta_{\text{old}}}(\cdot \mid q)$ for each prompt $q$. This formulation naturally handles reward scaling but encounters difficulties when all responses yield identical rewards (zero variance), resulting in undefined advantages that provide no training signal—a critical challenge for optimization.

### 3.2 OPPORTUNITIES FOR BETTER EXPLORATION-EXPLOITATION

Existing advances in GRPO employ a static strategy for both rollouts and reward assignment, uniformly distributing computational resources across prompts and assigning sparse rewards across answers. This rigid approach creates two key inefficiencies: insufficient exploration of valuable prompts and inadequate exploitation of the trajectories of generated answers. We next identify specific limitations in each dimension that motivate our work.

**Exploring uncertain, edge-of-policy prompts.** Allocating an equal generation budget across all prompts holds back the potential of edge-case prompts with high reward variance within the group, which produce steeper advantage signals. On the other hand, for complex problems that consistently yield zero rewards, this creates persistent learning bottlenecks: they provide no gradient signal, yet mastering these prompts is likely critical for pushing the boundary of model capability. To break this symmetry and perform effective exploration for these hard problems, we can leverage the idea of In-Context Learning (ICL) to introduce an ICL seeding strategy, which temporarily expands

the search space and uncovers strategies otherwise inaccessible—especially under their scarcity of correct cases, which are critical for guiding refinement. As shown in Figure 1a, applying ICL raises the accuracy from 37.7% to 41.5% for Qwen3-1.7B, and Figure 1b shows it flips and corrects 15%–20% of zero-accuracy prompts. These results demonstrate that ICL significantly enlarges the exploration space and improves generalization to harder problems.

**Exploiting trajectory differences in rewards.** The rule-based reward in GRPO assigns rollouts with identical advantage despite their differing strategies, leading to sparse or even degenerated learning signals. Prior methods tried to resolve through tree-structured sampling (Hou et al., 2025; Yang et al., 2025b) or data selection (Yu et al., 2025) but often require major architectural changes and overlook inherent features embedded in the generation process. We hypothesize that correct solutions with relatively low likelihood compared to other rollouts can drive the most effective learning in LLM reasoning, thus reasonably performing a novelty-guided advantage shaping. These rare successes represent critical learning opportunities: they expand the model's solution repertoire and prevent premature convergence to suboptimal but familiar patterns.

## 4  *XRPO* METHOD

*XRPO* creatively extends the capabilities of GRPO from an exploration–exploitation perspective. To encourage exploration on valuable prompts, we allocate the rollout budget for a given batch of prompts across multiple rounds, tackling the trade-off between estimated uncertainty reduction and exploration on sparsely sampled prompts. After these phased generations, we perform novelty-based advantage sharpening on all correct rollouts, promoting exploitation of rich trajectory signals.

### 4.1  EXPLORATION ON VALUABLE PROMPTS

The GRPO design suffers from two key ineffectiveness: (i) *degenerate reward groups*, where all $G$ responses are homogeneous (all passing or all failing), yielding a standard deviation of zero and thus an unusable advantage signal; and (ii) *myopic allocation of computational resources*, where rollouts are disproportionately spent on prompts with already well-established reward statistics, while ambiguous or underexplored prompts receive insufficient attention. Both issues stem from ignoring the inherent *uncertainty* in rollout sampling: each rollout is only a stochastic draw from the latent token distribution, and the quality of the estimated reward statistics depends heavily on how sampled rollouts are allocated across prompts.

**Hierarchical Rollout Planning.** To address GRPO's myopic allocation, an effective allocation algorithm should therefore satisfy three criteria: (1) *uncertainty-awareness*, prioritizing rollouts that most reduce statistical error; (2) *generalizability*, remaining agnostic to specific model architectures or reward scales; and (3) *lightweight implementation*, avoiding costly optimization in training.

We propose a novel hierarchical rollout strategy that models the priority of each prompt as a combination of both the expected reduction in uncertainty and an exploration bonus. Our design first prioritizes prompts with the best estimated reward uncertainty reduction. The uncertainty in the estimated mean reward $\mu_q$ can be quantified by the half-width of the Student's $t$-confidence interval:

$$h_q(n_q) = t_{1-\alpha/2, n_q-1} \frac{s_q}{\sqrt{n_q}}, \tag{1}$$

where $\bar{r}_q$ and $s_q$ are the sample mean and standard deviation of rewards for prompt $q$, $n_q$ is the number of rollouts, and $t_{1-\alpha/2,n_q-1}$ is the critical value of the $t$-distribution with $n_q - 1$ degrees of freedom at confidence $1 - \alpha$. Now the expected uncertainty reduction from one additional rollout can be approximated by

$$\hat{\Delta}_q(n_q) = h_q(n_q) - h_q(n_q + 1) \approx s_q \left( \frac{t_{1-\alpha/2,n_q-1}}{\sqrt{n_q}} - \frac{t_{1-\alpha/2,n_q}}{\sqrt{n_q+1}} \right). \tag{2}$$

This term favors prompts where an additional rollout provides the most statistical reward benefit.

However, simply allocating more rollout budgets to the uncertain prompts will overlook the sparsely sampled and hard prompts. Hence, we shift some amount of rollout budgets toward sparsely sampled

prompts by adding an exploration bonus, encouraging better exploration–exploitation trade-offs: $\phi_q(T, n_q) = \lambda \sqrt{\frac{\log(1+T)}{n_q}}$, where $T$ is the total number of rollouts allocated in the current round, and $\lambda > 0$ is a tunable hyperparameter that trades off uncertainty-driven exploitation against exploration.

As such, the final priority score for allocating the next rollout to prompt $q$ is

$$\Pi_q = \hat{\Delta}_q(n_q) + \phi_q(T, n_q). \tag{3}$$

To prevent cold-start issues and degenerate groupings, we adopt a phased rollout allocation strategy. Each prompt first receives $n_{\text{base}}$ rollouts to establish a baseline signal. For example, under a total budget of 128 rollouts, we partition the allocation into three rounds: the first 64 are uniformly distributed across prompts, while the remaining 64 are divided across the second and third phases, where each prompt receives rollouts in proportion to its current priority score $\Pi_q$. This phased design enables periodic re-estimation of both $\bar{r}_q$ and $s_q$, thereby stabilizing allocation dynamics over time. Consequently, our design preserves a balanced trade-off between reducing uncertainty for high-variance prompts and sustaining exploration on undersampled ones.

**Breaking Symmetry via ICL Seeding.** While our rollout planner primarily prioritizes high-variance prompts, hard prompts often continue to yield zero rewards, a prevalent phenomenon in GRPO (§3.2). Such systematically unsolved groups therefore require an additional mechanism to break the zero-reward symmetry.

Because these problems often lie far beyond the model's capacity, self-refinement methods based solely on its own responses (Ding et al., 2025) is hard to be effective. We therefore introduce the idea of ICL into GRPO training: conditioning on verified successes from similar tasks provides contextual guidance that breaks the zero-reward symmetry and enables in-context policy improvement even with fixed $\theta$. We formulate our ICL seeding strategy as follows.

For any prompt $q$ whose rollouts have all failed, we retrieve up to $K$ similar training questions with verified reward solutions from an evolving ICL corpus and build a compact few-shot template (Refer Appendix E). During rollout allocation phases, prompts with no successes spend their rollout budget on ICL seeding, while others receive standard rollouts.

## 4.2 EXPLOITATION OF THE SAMPLING TRAJECTORIES

While our rollout planner encourages exploration of valuable prompts, we observe that rule-based rewards (e.g., binary 0/1 signals) remain too sparse, often collapsing distinctions among diverse rollouts. This suppresses the rich information embedded in generation trajectories and drives the model toward homogenized, suboptimal behaviors. To overcome this limitation, we extend the classical entropy bonus (Williams, 1992; Mnih et al., 2016) from the token level to the sequence level, and instantiate it with a group-relative, novelty-aware advantage sharpening mechanism.

**Novelty-Guided Advantage Sharpening.** The concept of *novelty* could be intuitively interpreted as the extent to which a rollout deviates from the estimated entropy of the entire sequence trajectory space. Formally, in autoregressive models, token-level entropy is defined as $H_t = -\sum_{v \in \mathcal{V}} \pi_\theta(v \mid x, y_{<t}) \log \pi_\theta(v \mid x, y_{<t})$, while sequence-level entropy considers full trajectories: $H(\pi_\theta) = -\sum_{y \in \mathcal{Y}} \pi_\theta(y \mid x) \log \pi_\theta(y \mid x)$. Under autoregressive factorization, this becomes $H(\pi_\theta) = -\mathbb{E}_{y \sim \pi_\theta}\left[\sum_{t=1}^{|y|} \log \pi_\theta(y_t \mid x, y_{<t})\right]$. To estimate it in practice, we can define the length-normalized log-likelihood score for a sampled trajectory $y$ as

$$s(y) = \frac{1}{|y|} \sum_{t=1}^{|y|} \log \pi_\theta(y_t \mid x, y_{<t}), \tag{4}$$

and estimate the full trajectory space entropy with $|H(\pi_\theta)| \propto \frac{1}{N} \sum_{j=1}^{N} s(y_j) = \bar{s}$, which is the averaged length-normalized log-likelihood score $\bar{s}$ across $N$ sampled rollouts. Then the *novelty* of rollout $y_i$ is defined as

$$\eta_i = e^{s(y_i) - \bar{s}}, \tag{5}$$

---

**Algorithm 1** *XRPO*: eXplore-eXploit GRPO

---

**Require:** LLM $\pi_\theta$, evaluator Eval, base rollouts $n_{\text{base}}$, per-round allocation $n_r$, planning rounds
$N_{\text{plan}}$, batch $\mathcal{Q} = \{q\}$, ICL corpus $\mathcal{C}_{\text{init}}$
**Ensure:** Completed rollout set $Y_{\text{complete}}$; Advantage $A'$ and updated ICL corpus $\mathcal{C}$
1: $Y_{\text{complete}} \leftarrow \varnothing; \quad A' \leftarrow \varnothing; \quad \mathcal{C} \leftarrow \mathcal{C}_{\text{init}}$          ▷ initialize containers
2: **for** $q \in \mathcal{Q}$ **do**
3:     $Y_{\text{complete}}[q] \leftarrow \{\, y_{\text{base}}^{(i)} \sim \pi_\theta(\cdot \mid q) \,\}^{n_{\text{base}}}$          ▷ generate $n_{\text{base}}$ rollouts
4: **end for**

5: **for** $t = 1$ **to** $N_{\text{plan}}$ **do**          ▷ plan additional rollouts
6:     $S_\mathcal{Q} \leftarrow \{\text{Eval}(Y_{\text{complete}}[q])\}_{q \in \mathcal{Q}}$
7:     $\text{Alloc} \leftarrow \text{STATALLOCWITHEXPLORATION}(S_\mathcal{Q}, n_r)$
8:     **for** $q \in \mathcal{Q}$ **do**
9:        $\text{acc}(q) \leftarrow \mathbb{I}\{\exists\, y \in Y_{\text{complete}}[q] : \text{Eval}(y) = 1\}$     ▷ choose ICL iff no correct rollout
10:       $q_{prompt} \leftarrow \text{ICLPROMPT}(q, \mathcal{C})$ **if** $\text{acc}(q) = 0$ **else** $q$
11:       $Y_t \leftarrow \{\, y \sim \pi_\theta(\cdot \mid q_{prompt}) \,\}^{\text{Alloc}[q]}$          ▷ generate $\text{Alloc}[q]$ rollouts
12:       $Y_{\text{complete}}[q] \leftarrow Y_{\text{complete}}[q] \cup Y_t$
13:     **end for**
14: **end for**

15: $A^+ \leftarrow \text{ADVANTAGESHARPENING}(\text{Eval}, Y_{\text{complete}}[q], \pi_\theta)$       ▷ compute advantage
16: $\text{UPDATECORPUS}(\mathcal{C}, Y_{\text{complete}})$          ▷ add only correct, non-ICL rollouts
17: **return** $Y_{\text{complete}}, A', \mathcal{C}$

---

where $\eta_i < 1$ indicates a more uncertain (i.e., novel) sequence relative to the group. This provides a direct, sequence-level measure of how atypical a rollout is under the model's own distribution.

We integrate this *novelty* into training by sharpening the advantage for rule-based rewards. Specifically, if rollout $y_i$ receives full reward (e.g., 1), we adjust its default GRPO advantage $A_i$ as,

$$\text{A}_i^+ = \text{A}_i + \min\{\max\{\lambda_{\text{novelty}}(1 - \eta_i), 0\},\ \kappa_{\text{clip}} \cdot A_i\}. \tag{6}$$

where $\lambda_{\text{novelty}}$ controls how strong the novelty bonus is and $\kappa_{\text{clip}}$ caps the maximum bonus. Only rollouts with $\eta_i < 1$ (Refer Eq. 5) are boosted. Our sharpening mechanism offers two benefits: (i) *Boundary Expansion*: By rewarding novel yet correct sequences, the policy boundary is pushed outward, improving generalization. (ii) *Dense Differentiation*: Groups with degenerated advantages gain additional reward signals, mitigating collapse and accelerating convergence.

Algorithm 1 summarizes how *XRPO* integrates exploration and exploitation in a cohesive loop. After initializing with a fixed number of base rollouts per prompt, *XRPO* proceeds in phased allocation rounds (Lines 5–14), where rollout allocations are distributed by jointly considering the expected reduction in statistical uncertainty and the need to continue sampling underexplored prompts. For prompts that consistently fail, *XRPO* activates ICL seeding by injecting contextual examples from the evolving corpus (Line 10), breaking zero-reward symmetry and enabling policy improvement. At the end of each phase, the algorithm collects new rollouts, updates prompt-level statistics, and repeats. Once all rollouts generation is complete we compute advantage and correct rollouts are further refined with novelty-guided advantage sharpening.

## 5 EXPERIMENT

### 5.1 EXPERIMENTAL SETTINGS

Our method is implemented using the VERL pipeline (Sheng et al., 2024) and leverages vLLM (Kwon et al., 2023) for rollout execution. All experiments are conducted on $16\times$ H200 GPUs (141GB each).

**Models & Datasets.** We evaluate our approach on Qwen3-1.7B (Yang et al., 2025a). Qwen2.5-7B-Instruct (Qwen et al., 2024) and Llama-3.2-3B (Grattafiori et al., 2024). Following existing

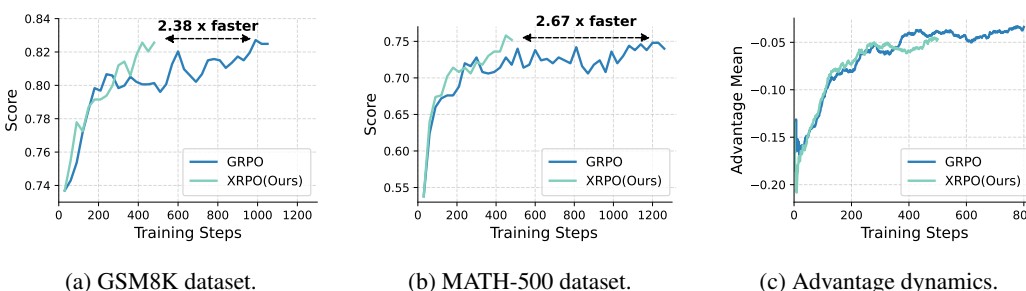

Figure 2: *XRPO* achieves faster training convergence than the baseline.

advances (Zheng et al., 2025a), Qwen3-1.7B is configured with a maximum generation length of 16,384 tokens and an input context window of 8,192 tokens, with reasoning mode enabled. For Qwen2.5-7B-Instruct, we use a maximum generation length of 8,192 tokens and a prompt length up to 4,096 tokens. For Llama-3.2-3B, we use a maximum generation length of 2,048 tokens and allow prompt lengths up to 4,096 tokens.

Our evaluations cover two primary tasks across seven datasets: (1) *Math Reasoning*: AIME 2024/2025 (AIME), HMMT 2025 (Feb) (Balunović et al., 2025), BRUMO 2025 (Balunović et al., 2025), and MATH (Lightman et al., 2023); and (2) *Code Generation*: Codeforces (Quan et al., 2025) and LiveCodeBench v5 (LCBv5) (Jain et al., 2024).

**Training Setup.**    We train using the AdamW optimizer with a learning rate of $2 \times 10^{-6}$, a batch size of 16, and a KL coefficient of $\beta = 0.001$. Our rollout strategy uses 4 base rollouts per prompt, with a total of 128 rollouts for Qwen3-1.7B and 64 for Qwen2.5-7B-Instruct, distributed across 2 dynamic rollout allocation rounds. For ICL seeding, we incorporate 2 solved exemplars generated by the same model and apply novelty-based advantage shaping with $\lambda_{\text{novelty}} = 2.5$ and $\kappa_{\text{clip}} = 0.5$. To construct the ICL corpus, problem similarity is computed using Qwen3-Embedding-8B (Zhang et al., 2025b). We further provide a sensitivity analysis to demonstrate *XRPO*'s consistent effectiveness (§5.3).

For all other experiments in the paper, we construct the training data by randomly sampling 10K examples from both the DAPO-Math-17k dataset (Yu et al., 2025) and the DeepCoder-Preview-Dataset (Luo et al., 2025), keeping the dataset size practical. For the training convergence experiment (Figure 2), we use Qwen2.5-Math-1.5B (Yang et al., 2024) with a maximum generation length of 2048 tokens and a maximum prompt length of 1024 tokens. This model is trained entirely on the MATH dataset (Lightman et al., 2023).

**Baselines.**    We compare against four state-of-the-art RL baselines: (1) GRPO (Shao et al., 2024), which applies token-level clipping with $\epsilon \in [0.2, 0.28]$, (2) GSPO (Zheng et al., 2025a), which optimizes at the sequence level with $\epsilon \in [0.0003, 0.0004]$, (3) Dynamic sampling strategy from DAPO (Yu et al., 2025), and (4) TreePO sampling strategy from TreePO (Li et al., 2025c), following the settings described in their respective papers.

**Evaluation Metrics.**    Following Wang et al. (2025a), we evaluate models every 30 training steps on the validation set. For inference, we generate $n = 32$ candidate solutions per problem using nucleus sampling (Temperature = 1.0, top-$p = 0.95$ for Qwen models and Temperature = 0.6, top-$p = 0.9$ for Llama-3.2-3B) and measure performance with `pass@k` (probability that at least one of the top-$k$ samples is correct) and `cons@32` (fraction of problems with correct majority-vote accuracy aggregated using 32 sampled solutions per problem.).

## 5.2 END-TO-END PERFORMANCE

We begin with an end-to-end analysis of efficiency and model quality to assess the effectiveness of our prompt exploration and reward exploitation strategies.

| Method | Metric | AIME'24 | AIME'25 | HMMT'25 | BRUMO'25 | MATH | Codeforces | LCBv5 | Avg. |
|--------|--------|---------|---------|---------|----------|------|------------|-------|------|
| | | | | | *Qwen3-1.7B* | | | | |
| GSPO | pass@1 | 42.39 | 33.23 | 21.25 | 45.20 | 90.33 | 13.72 | **33.83** | 39.99 |
| | cons@32 | **60.00** | 46.67 | 20.00 | 56.67 | – | – | – | 45.84 |
| DS | pass@1 | 40.31 | 32.50 | 20.42 | 39.90 | 89.33 | 9.48 | 29.81 | 37.39 |
| | cons@32 | 43.33 | 33.33 | 23.33 | 43.33 | – | – | – | 35.83 |
| TPO-S | pass@1 | 38.33 | 26.04 | 17.29 | 37.40 | 84.65 | 9.51 | 30.24 | 34.78 |
| | cons@32 | 50.00 | 33.33 | 20.00 | 46.67 | – | – | – | 37.50 |
| *XRPO* | pass@1 | **46.46** | **35.72** | **22.29** | **47.39** | **90.54** | **13.80** | 33.74 | **41.42** |
| | cons@32 | 56.67 | **50.00** | **26.67** | **60.00** | – | – | – | **48.34** |

Table 1: Comparison of GSPO, Dynamic Sampling (DS) and TreePO Sampling (TPO-S) with *XRPO* across benchmarks for Qwen3-1.7B (Reasoning).

| Method | Metric | AIME'24 | AIME'25 | BRUMO'25 | MATH-500 | Avg. |
|--------|--------|---------|---------|----------|----------|------|
| | | | *Llama-3.2-3B* | | | |
| GRPO | pass@1 | 7.19 | 0.52 | **4.79** | 43.04 | 13.88 |
| | pass@4 | 16.17 | 1.97 | 7.82 | **62.08** | 22.01 |
| *XRPO* | pass@1 | **9.27** | **0.63** | 3.54 | **43.40** | **14.21** |
| | pass@4 | **18.70** | **2.38** | **9.48** | 61.24 | **22.95** |

Table 2: Comparison of GRPO and *XRPO* across benchmarks for Llama-3.2-3B.

**XRPO significantly improves post-training model quality.** Table 1, Figure 3a and Table 2 show that *XRPO* consistently outperforms state-of-the-art baselines across challenging math reasoning and code generation benchmarks. For Qwen2.5-7B-Instruct, *XRPO* delivers substantial relative improvements of +9.2% in pass@4 and +20% in cons@32. For the complete results and comparisons with all baselines, please refer to Appendix C. On Llama-3.2-3B, AIME'25 remains extremely challenging and both *XRPO* and GRPO obtain low scores, yet *XRPO* still achieves a higher average performance despite the model's older architecture. Since this model is older, we also do not report const@32 because the scores were low for both methods. For Qwen3-1.7B (with reasoning mode enabled), *XRPO* achieves a +2.5% improvement in average cons@32 and +1.4% in pass@1 over GSPO across datasets, demonstrating gains in both raw accuracy and output consistency. This improvement is particularly notable given that GSPO is among the most recent and competitive approaches (Zheng et al., 2025a), with evaluations conducted on especially challenging datasets. Further, Qwen3 is already a reasoning-focused model and challenging to train due to its heavily trained nature effectively thus observing improvements highlights the practical value of *XRPO*. Moreover, *XRPO* not only enhances accuracy but also delivers superior training and inference efficiency (see later).

**XRPO achieves substantially faster training convergence.** Beyond accuracy improvements, Figure 2a shows that *XRPO* achieves noticeable training efficiency speedup on Qwen2.5-Math-1.7B when trained on the MATH training dataset. Specifically, it reaches 82.5% accuracy on GSM8K by step 420, whereas GRPO requires approximately 1K steps to achieve similar performance, yielding a speedup of $2.4\times$. Similarly, on MATH-500 (Lightman et al., 2023) (Figure 2b), *XRPO* attains 75% accuracy by step 450, compared to GRPO's about 1.2K steps, corresponding to a $2.7\times$ speedup. Figure 2c further shows that XRPO's advantage metric converges earlier, entering a more stable learning phase. These findings validate the efficiency of our exploration–exploitation design: by focusing rollout allocation on edge-policy cases, the model quickly absorbs salient information and effectively expands its decision boundaries.

**XRPO introduces negligible training overhead.** *XRPO* only introduces limited overhead and could operate at comparable latency compared to the vanilla GRPO algorithm. We evaluated the per-step end-to-end latency using a batch size of 64, 256 rollouts per prompt, and two dynamic rounds to better simulate common training settings. Our results indicate that the per-step latency ratio between *XRPO* and baseline GRPO is about 1.047, mere 4.7% overhead. The ICL corpus is loaded once at the start of training (4.22 seconds), advantage-shaping introduces nearly no overhead

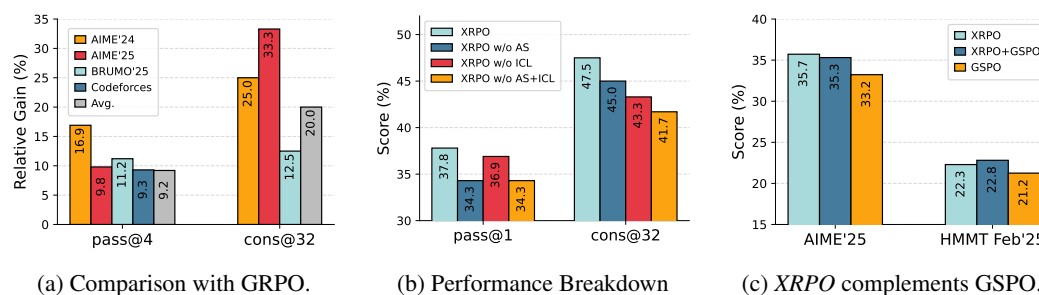

(a) Comparison with GRPO.  (b) Performance Breakdown  (c) *XRPO* complements GSPO.

Figure 3: *XRPO* improves performance across multiple dimensions.

(0.329 seconds), and constructing ICL prompts is extremely fast ($< 0.001$ seconds per prompt) due to simple hash-map lookups over precomputed neighbors and responses. Other than that, dynamic allocation planning takes 79.5 seconds, and updating the ICL corpus takes 11.8 seconds on average.

We further provide a theoretical latency analysis in Appendix D. This analysis shows that the overhead introduced by dynamic rollout allocation becomes negligible when the rollout workload is large, while remaining bounded in the worst case. Overall, *XRPO* not only reduces the number of convergence steps but also maintains near-identical per-step latency.

**XRPO achieves better inference efficiency.**   Figure 4a shows that *XRPO* reaches correct solutions with substantially shorter response lengths, reflecting more efficient and precise reasoning. In particular, *XRPO* reduces average response length by 13.6% on AIME'24 and 13.4% on AIME'25 relative to GRPO. This indicates that *XRPO* learns to reason in a more targeted fashion, avoiding redundant steps and converging to solutions more directly. These results further support the analysis presented in Appendix F.

### 5.3 ABLATION STUDY AND SENSITIVITY ANALYSIS

**Performance Breakdown by Design Components.**   We ablate *XRPO* into three variants to isolate the contribution of each component: (1) XRPO *w/o ICL*, which removes ICL Seeding (no symmetry-breaking); (2) XRPO *w/o AS*, which disables Advantage Sharpening (no trajectory-level adjustment); and (3) XRPO *w/o (AS+ICL)*, which removes both ICL Seeding and Advantage Sharpening, leaving rollout planning as the sole training signal.

Figure 3b summarizes results on Qwen3-1.7B (reasoning mode) averaged across AIME'24, AIME'25, HMMT'25, and BRUMO'25. Removing any module consistently degrades performance, highlighting their integrity to *XRPO*. Dropping ICL causes a 4.2% decline in cons@32 performance, highlighting the importance of symmetry-breaking. Disabling AS reduces sample efficiency and weakens generalization, leading to 3.5% accuracy degradation in pass@1, confirming the value of trajectory-level signals. Finally, while rollout planning alone provides a strong baseline, only the full combination of RP, ICL, and AS delivers the best performance, showing that all three components are necessary to fully realize the benefits of *XRPO*. For dataset-wise comparison, see Appendix B.

We further analyze how ICL contributes to better model convergence. Figure 4b shows that with ICL, *XRPO* correctly flips hard questions by an average of 6.2% and 4.2% on Qwen2.5-7B-Instruct and Qwen3-1.7B, respectively, demonstrating its effectiveness in breaking the edge of model capability. Similarly, Figure 4c indicates that ICL largely reduces the fraction of prompts that fail to produce complete responses within the 16K context length across all rollouts, typically complex, multi-step reasoning questions that otherwise confuse the model. These directly illustrate ICL's role in enabling the model to tackle problems beyond its decision boundary.

**Robustness to Hyperparameters.**   To evaluate the robustness of our method to hyperparameter choices, we conduct experiments on Qwen2.5-1.5B with results MATH-500 dataset(Lightman et al., 2023) by varying both the novelty bonus $\lambda_{\text{novelty}}$ and the clamp factor $\kappa_{\text{clip}}$, , as well as the hyperparameters related to hierarchical rollout planning, including the exploration strength $\lambda$ and the confidence interval $\alpha$. Results are shown respectively in Table 3a, and Table 3b. Overall, performance

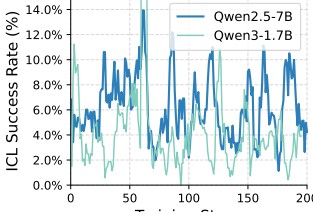
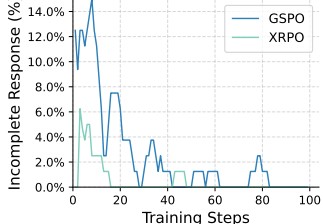

(a) *XRPO* achieves better inference efficiency.

(b) *XRPO* maintains a consistently stable ICL success rate.

(c) *XRPO* achieves better response completion rates.

Figure 4: Break down analysis on *XRPO*.

| $\lambda_{\text{novelty}}$ | $\kappa_{\text{clip}}$ | Score (%) |
|:---:|:---:|:---:|
| 2.5 | 0.5 | 53.66 |
| 5.0 | 0.5 | 54.94 |
| 1.0 | 0.5 | 56.26 |
| 2.5 | 0.8 | 55.24 |
| average | | $55.02 \pm 1.07$ |

(a) Novelty bonus $\lambda_{\text{novelty}}$ and clamp factor $\kappa_{\text{clip}}$.

| $\lambda$ | $\alpha$ | Score (%) |
|:---:|:---:|:---:|
| 0.10 | 0.95 | 53.66 |
| 0.12 | 0.93 | 53.06 |
| 0.08 | 0.95 | 56.27 |
| 0.12 | 0.97 | 55.32 |
| 0.08 | 0.93 | 55.89 |
| average | | $54.84 \pm 1.41$ |

(b) Exploration strength $\lambda$ and confidence $\alpha$.

Table 3: Hyperparameter sensitivity analysis.

remains stable across a broad range of settings. When varying $\lambda_{\text{novelty}}$ and the clamp factor $\kappa_{\text{clip}}$, accuracies fluctuate only slightly, with a mean of 55.02% and a sample standard deviation of 1.07. As for $\lambda$ and $\alpha$, the accuracies fluctuate within a narrow band with a mean of 54.84% and a sample standard deviation of 1.41. These findings confirm that our design is resilient to hyperparameter variations, ensuring reliable performance without requiring extensive tuning.

**Compatibility with Other SOTA Methods.** Our design is complementary to recent optimization advances and can be seamlessly integrated with state-of-the-art methods. As shown in Figure 3c, *XRPO* achieves 2.5% better accuracy than GSPO when applied on Qwen3-1.7B. Importantly, *XRPO* remains fully compatible and yields comparable or improved performance when paired with GSPO. These results highlight that our exploration–exploitation mechanisms complement GSPO's optimization strategy, further enhancing downstream reasoning quality.

# 6 CONCLUSION

This paper introduces *XRPO*, a principled RLHF framework that rebalances exploration and exploitation in rollout optimization. By introducing hierarchical rollout planning that prioritizes high-variance prompts near decision boundaries, ICL seeding that breaks zero-reward symmetry on hard problems, and novelty-aware advantage sharpening that amplifies low-probability yet correct responses, XRPO pushes models beyond their current capability limits. Extensive experiments demonstrate consistent improvements over GRPO and recent advances, with over 1.4% higher accuracy, 2.5% higher average consistency across math and coding benchmarks, and 2.7× faster convergence.

# 7 REPRODUCIBILITY STATEMENT

To ensure the reproducibility of our work, we provide a clear description of our methods in Section 4, including detailed mathematical formulations and the algorithm pseudocode in Algorithm 1. All experimental settings are reported in Section 5.1, using entirely publicly available datasets (e.g., AIME 2024/2025 (AIME), BRUMO 2025 (Balunović et al., 2025)) and open-source models (Qwen3-1.7B

and Qwen2.5-7B-Instruct). We also share the in-context learning (ICL) prompts used in our experiments in Appendix E and make our implementation available in the supplementary material.

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

APPENDIX

## A  THE USE OF LARGE LANGUAGE MODELS (LLMS)

LLMs were used solely for minor editorial assistance (grammar and phrasing) and for routine software support (code refinement, automating repetitive tasks). They were not used for research ideation, data analysis, or substantive writing. All outputs were reviewed and verified by the authors, and LLMs do not meet authorship or contributor criteria.

## B  DATASET-WISE COMPARISON OF *XRPO* COMPONENTS AND BASELINES.

The results in Table 4 show that the ablated XRPO variants consistently outperform the other sampling baselines. Even when AS, ICL, or both are removed, these reduced versions still achieve higher pass@1 and const@32 scores than Dynamic Sampling and TreePO Sampling across most datasets. This indicates that the underlying XRPO framework remains strong even without its individual components.

| Method | Metric | AIME'24 | AIME'25 | HMMT'25 | BRUMO'25 | Avg. |
|---|---|---|---|---|---|---|
| | | *Qwen3-1.7B* | | | | |
| XRPO | pass@1 | **46.46** | 35.72 | 21.66 | **47.39** | **37.81** |
| | const@32 | **56.66** | **50.00** | **23.33** | **60.00** | **47.50** |
| XRPO w/o AS | pass@1 | 39.17 | 33.13 | 20.10 | 45.00 | 34.35 |
| | const@32 | 53.33 | 40.00 | **26.67** | 60.00 | 45.00 |
| XRPO w/o ICL | pass@1 | 44.58 | **35.94** | 20.94 | 46.04 | 36.87 |
| | const@32 | 53.33 | 43.33 | 20.00 | 56.67 | 43.33 |
| XRPO w/o AS+ICL | pass@1 | 40.72 | 31.45 | 19.06 | 46.04 | 34.32 |
| | const@32 | 46.66 | 40.00 | 20.00 | 60.00 | 41.67 |
| Dynamic Sampling | pass@1 | 40.31 | 32.50 | 20.42 | 39.90 | 33.28 |
| | const@32 | 43.33 | 33.33 | 23.33 | 43.33 | 35.83 |
| TreePO Sampling | pass@1 | 38.33 | 26.04 | 17.29 | 37.40 | 29.77 |
| | const@32 | 50.00 | 33.33 | 20.00 | 46.67 | 37.50 |

Table 4: Dataset-wise comparison.

## C  COMPARISON OF *XRPO* AND BASELINES ON QWEN2.5-7B-INSTRUCT

In Table 5 we present expanded evaluation results for Qwen2.5-7B-Instruct across the AIME and BRUMO benchmarks.

## D  LATENCY ANLYSIS ON *XRPO*

Let $N$ denote the total rollout budget, $m_p$ the system parallelism, and $t_0$ the per-rollout execution time. The baseline uniform allocation requires

$$T_1 = \left\lceil \frac{N}{m_p} \right\rceil t_0 = \frac{N + m_p - 1}{m_p} t_0. \tag{7}$$

Assuming we set the dynamic rounds to $n$, which divides $N$. Including the planning overhead per-round, $\Delta t$, the total runtime will be

$$T_2 = n\left( \left\lceil \frac{N/n}{m_p} \right\rceil t_0 + \Delta t \right) = \frac{N + nm_p - n}{m_p} t_0 + n\Delta t. \tag{8}$$

| Method | Metric | AIME'24 | AIME'25 | BRUMO'25 | Avg. |
|---|---|---|---|---|---|
| | | *Qwen2.5-7B-Instruct* | | | |
| GRPO | pass@1 | 10.31 | 6.73 | 15.83 | 10.96 |
| | pass@4 | 18.10 | 15.31 | 26.30 | 19.90 |
| | const@32 | 13.33 | 10.00 | 26.67 | 16.67 |
| Dynamic Sampling | pass@1 | 10.52 | 6.67 | 17.92 | 11.70 |
| | pass@4 | 20.22 | 15.57 | 28.72 | 21.50 |
| | const@32 | 16.67 | 10.00 | 26.67 | 17.78 |
| TreePO Sampling | pass@1 | 8.96 | 5.10 | **19.17** | 11.08 |
| | pass@4 | 17.20 | 13.72 | **29.55** | 20.16 |
| | const@32 | 16.67 | 6.67 | 26.67 | 16.67 |
| XRPO | pass@1 | **11.25** | **7.71** | 17.19 | **12.05** |
| | pass@4 | **21.17** | **16.80** | 29.25 | **22.41** |
| | const@32 | **16.67** | **13.33** | **30.00** | **20.00** |
| Rel. Gain wrt GRPO. | pass@1 | 9.09% | 14.55% | 8.55% | 10.73% |
| | pass@4 | 16.96% | 9.73% | 11.22% | 12.64% |
| | const@32 | 24.98% | 33.30% | 12.53% | 23.60% |

Table 5: Results on Qwen2.5-7B.

Since $\Delta t$ involves only lightweight statistics (e.g., computing reward variances), it is negligible compared to rollout execution. The latency ratio can be rewritten explicitly as

$$\frac{T_2}{T_1} = \frac{N + nm_p - n}{N + m_p - 1} = n - \frac{(n-1)N}{N + m_p - 1} = n - \frac{(n-1)}{1 + \frac{m_p - 1}{N}} \tag{9}$$

If the total batch size $N$ is far greater than the degree of parallelism $m_p$ (*i.e.* $\lim \frac{m_p}{N} \to 0$), the dynamic allocator introduces no overhead. It is shown as

$$\frac{T_2}{T_1} = n - (n-1) = 1 \tag{10}$$

If we consider the worst case when $N$ is significantly small compared to $m_p$(*i.e.* $\lim \frac{m_p}{N} \to \infty$), which is very unlikely to happen, we show that $T_2$ is still bounded as

$$\frac{T_2}{T_1} = n - 0 = n \tag{11}$$

Therefore, we could safely conclude that the overhead for dynamic allocation is negligible for large $N$, which is the typical case for both training and deployment, while still remaining bounded in the worst case.

# E  BREAKING SYMMETRY VIA ICL SEEDING

To address the challenge of systematically unsolved prompts, we integrate a few-shot in-context learning (ICL) seeding strategy into *XRPO* training. Whenever a given prompt fails to produce any successful solution in its base rollouts, the remaining rollout budget is allocated to an ICL-augmented prompt. These ICL prompts incorporate verified solved examples drawn from an evolving corpus of problem–solution pairs that the model has successfully answered in prior training steps. The similarity search for retrieval is conducted using Qwen3-Embedding-8B (Zhang et al., 2025b), ensuring that only the most semantically relevant solved problems are selected as demonstrations. We limit the number of retrieved examples to $K = 2$ in order to conserve context length while still providing sufficient guidance. If no suitable solved examples exist, the prompt falls back to its zero-shot form.

| Metric | w.r.t. Correct Rollouts | w.r.t. Full Groups |
|---|---|---|
| Z-score | 0.257 | -0.280 |
| Relative Ratio | 1.040 | 0.991 |

Table 6: Length statistics of shaped entries relative to correct rollouts and full groups.

The prompt template is structured into three components: (i) a `<task>` section containing general instructions, (ii) an `<examples>` block containing up to two similar problems and their corresponding verified solutions, and (iii) the new problem to be solved, formatted within a `<new_problem>` tag. The model is instructed to extract a general strategy from the examples, reason through the new problem, and finally output the answer in a standardized format (e.g., \boxed{} for mathematics or fenced code blocks for programming). To ensure feasibility within the model's context window, overly long example solutions are truncated as needed.

**Prompt Template.** ICL prompt used in our experiments is shown below:

```
ICL Prompt Template

<task>
  You are given several worked examples, each with a
  <problem> and a <solution>. Extract a general strategy,
  then think through the new problem, and finally provide
  the detailed solution.
</task>

<examples>
  <example id="1">
    <problem>[Example problem 1]</problem>
    <solution>[Correct solution 1]</solution>
  </example>

  <example id="2">
    <problem>[Example problem 2]</problem>
    <solution>[Correct solution 2]</solution>
  </example>
</examples>

<new_problem>[Hard unsolved problem]</new_problem>
```

## F    EFFECT OF NOVELTY-GUIDED ADVANTAGE SHARPENING AND RESPONSE LENGTH BEHAVIOR

### F.1   NOVELTY-GUIDED ADVANTAGE SHARPENING IS FREE OF LENGTH BIAS

We analyze whether the novelty-guided advantage sharpening mechanism introduces any preference for longer or shorter responses. Two observations confirm that the mechanism is free of such bias.

First, Equation 4 shows that each trajectory's log-likelihood score is normalized by its own length. This removes systematic preference for either long or short trajectories.

Second, we examine the length distribution of shaped entries by computing their response-length z-scores and their relative ratios within full groups and within the sets of correct rollouts in those groups (Table 6). The results indicate that shaped entries lie within 0.25 standard deviations of the mean and are more than 99 percent close to the group-average length. These findings show no observable length bias introduced by the novelty score or the shaping mechanism.

| Response Length | Qwen2.5 | Qwen3 |
|---|---|---|
| Without ICL | 881.7 | 12906.1 |
| With ICL | 827.2 | 8427.9 |
| Reduction | 6.17% | 34.7% |

Table 7: ICL reduces average generation length for both Qwen3 and Qwen2.5.

## F.2 XRPO LEARNS MORE EFFICIENT AND CONCISE RESPONSE PATTERNS

Although XRPO does not include an explicit length penalty, it consistently generates shorter and more efficient solution paths. This behavior emerges naturally from improved exploration within context constraints.

GRPO often exhibits length bias, producing long and repetitive reasoning chains. On difficult problems, these chains frequently exceed the model's maximum context window, causing truncation and yielding zero reward. GRPO therefore struggles to learn from such failed trajectories.

XRPO avoids this failure mode. The Rollout Allocator and In-Context Seeding guide the model toward solution paths that are likely to remain valid within the context window. Because successful solutions must fit within that window, XRPO implicitly favors trajectories that are concise enough to be complete. This interpretation is supported by empirical findings in Table 7, which show that the use of in-context learning reduces average generation length by 34.7 percent for Qwen3 and 6.17 percent for Qwen2.5.

## G  QUALITATIVE ANALYSIS

In this section, we present examples from the AIME 2025 and HMMT 2025 competition sets to illustrate the practical differences between GSPO and XRPO. In Example 1, drawn from the AIME 2025 set, GSPO arrives at an inflated value of $n + t = 159$ by applying a uniform counting strategy, whereas XRPO carefully distinguishes boundary cases ($k = 0$, $k = \pm 7$), avoids double counting, and correctly filters out invalid tangency conditions, yielding the correct solution $n + t = 149$. This methodological precision translates into measurable improvements on this problem XRPO achieves a pass@1 accuracy of 60% compared to only 40% for GSPO.

---

**Example1: Problem from AIME 2025**

There are $n$ values of $x$ in the interval $0 < x < 2\pi$ where

$$f(x) = \sin\bigl(7\pi \cdot \sin(5x)\bigr) = 0.$$

For $t$ of these $n$ values of $x$, the graph of $y = f(x)$ is tangent to the $x$-axis. Find $n + t$. Let's think step by step and output the final answer within $\Box$.

---

---

### Example 1: GSPO

To solve the given problem, we analyze the function:

$$f(x) = \sin\left(7\pi \cdot \sin(5x)\right).$$

We are asked to determine the number of values of $x$ in the interval $0 < x < 2\pi$ for which $f(x) = 0$, and then find how many of these are points where the graph is tangent to the $x$-axis. Finally, we compute $n + t$, where $n$ is the total number of such points and $t$ is the number of those points where the graph is tangent.

**Step 1: Determine when $f(x) = 0$.**

$$f(x) = \sin\left(7\pi \cdot \sin(5x)\right) = 0$$

This happens when the argument of the sine function is an integer multiple of $\pi$:

$$7\pi \cdot \sin(5x) = k\pi \quad \Rightarrow \quad \sin(5x) = \frac{k}{7}.$$

Given that $\sin(\theta) \in [-1, 1]$, it follows that $\frac{k}{7} \in [-1, 1]$, so $k \in \{-7, -6, \ldots, 0, \ldots, 6, 7\}$, for a total of 15 possible values of $k$.

For each $k \neq 0$, the equation $\sin(5x) = \frac{k}{7}$ has *two* solutions in each $2\pi$ interval. Since $5x \in (0, 10\pi)$, which is five full $2\pi$ intervals, we find:

- For $k \neq 0$: 10 solutions each,

- For $k = 0$: 9 solutions (as $\sin(5x) = 0$ has 9 solutions in $(0, 10\pi)$).

Thus, $n = 14 \times 10 + 9 = 149$.

**Step 2: Determine where the graph is tangent to the $x$-axis.**
A function is tangent to the $x$-axis at a point if both $f(x) = 0$ and $f'(x) = 0$.
We compute the derivative:

$$f'(x) = \frac{d}{dx}\left[\sin\left(7\pi \cdot \sin(5x)\right)\right] = 35\pi \cdot \cos\left(7\pi \cdot \sin(5x)\right) \cdot \cos(5x).$$

This derivative is zero when either: $\cos\left(7\pi \cdot \sin(5x)\right) = 0$, or $\cos(5x) = 0$

**Step 3: Find points where $f(x) = 0$ and $f'(x) = 0$.**
We separately analyze the two cases.
*Case 1:* $\cos(5x) = 0$.

$$\cos(5x) = 0 \Rightarrow 5x = \frac{\pi}{2} + n\pi \Rightarrow x = \frac{\pi}{10} + \frac{n\pi}{5}.$$

Restricting $0 < x < 2\pi$, valid values for $n$ are $0, 1, 2, \ldots, 9$ (a total of 10 solutions).
So, the 10 solutions are

$$x = \frac{\pi}{10}, \frac{3\pi}{10}, \frac{5\pi}{10}, \ldots, \frac{19\pi}{10}.$$

These values are *unique* and lie within $(0, 2\pi)$.
Now check if each of these satisfies $\sin(5x) = \frac{k}{7}$ for some integer $k \neq 0$. Compute $5x$:

$$5x = \frac{\pi}{2} + n\pi \Rightarrow \sin(5x) = \sin\left(\frac{\pi}{2} + n\pi\right) = \cos(n\pi) = (-1)^n.$$

Thus,

$$\sin(5x) = \pm 1 \Rightarrow \frac{k}{7} = \pm 1 \Rightarrow k = \pm 7.$$

So, *each of these 10 points* satisfies $f(x) = 0$, and they are all valid solutions.

**Step 4: Compute $n + t$.**

$$n = 149, \qquad t = 10 \text{ (number of points where } f(x) = 0 \text{ and } f'(x) = 0\text{)}.$$

Therefore,

$$n + t = 149 + 10 = \boxed{159}.$$

---

---

### Example 1: XRPO

**Understanding the Problem**

We are given the function:

$$f(x) = \sin\left(7\pi \cdot \sin(5x)\right)$$

and asked to find the number of values of $x$ in the interval $0 < x < 2\pi$ such that $f(x) = 0$, denote these as $n$, and then determine how many of these $n$ values have the graph of $f(x)$ **tangent** to the $x$-axis (i.e., $f'(x) = 0$), denoted as $t$. The problem then requires us to compute $n + t$.

**Step 1: Solving $f(x) = 0$**

We solve the equation: $\sin\left(7\pi \cdot \sin(5x)\right) = 0$

This implies: $7\pi \cdot \sin(5x) = k\pi$ for some integer $k \Rightarrow \sin(5x) = \frac{k}{7}$.

Thus, $\sin(5x) = \frac{k}{7}$, where $k \in \{-7, -6, \ldots, 0, \ldots, 6, 7\}$, because $\frac{k}{7} \in [-1, 1]$.

This gives an equation for $x$ in terms of the integer $k$, and the total number of such $x$ values in the interval $(0, 2\pi)$ is determined by the number of solutions to $\sin(5x) = \frac{k}{7}$ in that domain.

**Step 2: Counting Number of Solutions $n$**

We analyze the number of solutions for each $k$:

*Case 1: $k = 0$*

$$\sin(5x) = 0 \Rightarrow 5x = n\pi \Rightarrow x = \frac{n\pi}{5}, \quad \text{for } n = 1, 2, \ldots, 9.$$

So there are **9** such solutions.

*Case 2: $|k| = 1$*

$$k = \pm 7 \Rightarrow \frac{k}{7} = \pm 1,$$

$$\sin(5x) = 1 \Rightarrow 5x = \frac{\pi}{2} + 2n\pi, \qquad \sin(5x) = -1 \Rightarrow 5x = \frac{3\pi}{2} + 2n\pi.$$

Each of these forms a periodic solution in $x$ on $(0, 2\pi)$, resulting in **5** solutions each. So for $k = \pm 7$, there are $5 + 5 = 10$ solutions.

*Case 3: $|k| < 1$*

$$k = -6, -5, \ldots, -1 \quad \text{and} \quad k = 1, 2, \ldots, 6 \quad \text{(12 values)}.$$

Each of these values has $\frac{k}{7} \in (-1, 1)$. Each such value of $\sin(5x)$ has **2** distinct solutions per period of $2\pi$, so over 5 full periods (since $\theta = 5x \in (0, 10\pi)$), the total is **10 solutions per** $k$.

Therefore, for 12 such $k$-values, we get $12 \times 10 = 120$ solutions.

Total Number of Solutions $n = 9$ (from $k = 0$) + 10 (from $k = \pm 7$) + 120 (other $k$ values) = 139.

**Step 3: Finding $t$ — Tangent Points**

A point is tangent to the $x$-axis if and only if $f(x) = 0$ and $f'(x) = 0$.

From the derivative:

$$f'(x) = \frac{d}{dx}\left[\sin(7\pi \sin(5x))\right] = 35\pi \cos(7\pi \sin(5x)) \cos(5x).$$

So $f'(x) = 0$ when: $\cos(5x) = 0$ or $\cos(7\pi \sin(5x)) = 0$.

*Case A: $\cos(5x) = 0$*

$$5x = \frac{\pi}{2} + n\pi \Rightarrow x = \frac{\pi}{10} + \frac{n\pi}{5}.$$

With $x \in (0, 2\pi)$, the valid $n$ are from 0 to 9, producing **10 distinct values**. Each such $x$ satisfies $f(x) = 0$, since $\sin(5x) = \sin\left(\frac{\pi}{2} + n\pi\right) = (-1)^n$. Thus, $k = \pm 7$, and these are valid.

*Case B: $\cos(7\pi \sin(5x)) = 0$*

$$7\pi \sin(5x) = \frac{\pi}{2} + m\pi \Rightarrow \sin(5x) = \frac{m + 1/2}{7}.$$

This would require $\sin(5x) = \frac{m+1/2}{7}$, but since $m \in \mathbb{Z}$, this introduces fractional values that cannot match the integer $k = \frac{m}{7}$ for $k \in \mathbb{Z}$. Therefore, no such $x$ can satisfy both $f(x) = 0$ and $f'(x) = 0$.

**Final Calculation**

$$n + t = 139 + 10 = \boxed{149}.$$

Example 2 is drawn from the HMMT 2025 set, the problem involves evaluating a challenging floor sum where GSPO applies a uniform pairing strategy and arrives at the incorrect final result of -1985, while XRPO carefully splits the sum into cases, distinguishes positive and negative indices, accounts for odd divisors of 4050, and arrives at the correct answer of -984. This refinement again highlights XRPO's superior reasoning rigor and ability to handle intricate floor and ceiling interactions, leading to measurable gains on this promblem XRPO achieves a pass@1 accuracy of 21.8% compared to only 9.37% for GSPO.

---

**Example 2: Problem from HMMT 2025**

Let $\lfloor z \rfloor$ denote the greatest integer less than or equal to $z$. Compute

$$\sum_{j=-1000}^{1000} \left\lfloor \frac{2025}{j + 0.5} \right\rfloor$$

Let's think step by step and output the final answer within $\square$.

## Example 2: GSPO

To compute the following sum:

$$\sum_{j=-1000}^{1000} \left\lfloor \frac{2025}{j+0.5} \right\rfloor$$

we begin by simplifying the expression inside the floor function.

**Step 1: Reduce the Expression**

We note that:

$$\frac{2025}{j+0.5} = \frac{2025}{\frac{2j+1}{2}} = \frac{4050}{2j+1}$$

Hence, the original sum can be rewritten as:

$$\sum_{j=-1000}^{1000} \left\lfloor \frac{4050}{2j+1} \right\rfloor$$

**Step 2: Change of Variables**

Let's make a substitution to simplify the range of $j$. Let $n = j+1$, then $2j+1 = 2(n-1)+1 = 2n-1$. Thus, the sum becomes:

$$\sum_{j=-1000}^{1000} \left\lfloor \frac{4050}{2j+1} \right\rfloor = \sum_{n=1}^{2001} \left\lfloor \frac{4050}{2n-1} \right\rfloor$$

This is valid for $n = 1, 2, \ldots, 2001$, since: When $j = -1000$, $2j+1 = -1999$ and for $j = 1000$, $2j+1 = 2001$. Thus, the total sum becomes:

$$\sum_{n=1}^{2001} \left\lfloor \frac{4050}{2n-1} \right\rfloor$$

**Step 3: Use Symmetry and Pairing**

The sum is a bit complex to directly evaluate due to the floor function and the range of $n$. However, we note that:

For any integer $k \in \mathbb{Z}$, $\left\lfloor \frac{4050}{k} \right\rfloor + \left\lfloor \frac{4050}{-k} \right\rfloor = \left\lfloor \frac{4050}{k} \right\rfloor + \left\lfloor \frac{4050}{-k} \right\rfloor$.

But since $\left\lfloor \frac{4050}{-k} \right\rfloor = -\left\lceil \frac{4050}{k} \right\rceil$, this becomes: $\left\lfloor \frac{4050}{k} \right\rfloor - \left\lceil \frac{4050}{k} \right\rceil$.

We denote $x = \frac{4050}{k}$ and observe that: If $x$ is an integer, then $\lfloor x \rfloor - \lceil x \rceil = 0$ and if $x$ is not an integer, then $\lfloor x \rfloor - \lceil x \rceil = -1$. So, for all $k$ such that $x = \frac{4050}{k}$ is not an integer, the contribution to the total sum is $-1$, and for those where $x$ is an integer, it contributes 0.

**Step 4: Count the Terms Where $\frac{4050}{k}$ is Not an Integer**

We first determine all odd integers $k = 2n - 1$, $n = 1, 2, \ldots, 2001$, such that $k$ divides 4050.

*Prime factorization of 4050:* $4050 = 2 \cdot 3^4 \cdot 5^2$. Since we are only interested in odd divisors (i.e., not including the 2), the odd divisors are the divisors of $3^4 \cdot 5^2 = 81 \cdot 25 = 2025$. The number of odd divisors of 2025 is: $(4+1)(2+1) = 15$. From these, we exclude the divisor 2025 itself (since $k = 2001$ is the largest $k$ allowed, but $2025 > 2001$), leaving **14** valid odd divisors less than or equal to 2001. So, among the 2001 terms in the range $n = 1$ to 2001, **14 of them** make $\frac{4050}{k}$ an integer.

**Step 5: Final Computation**

Each of the 14 terms where $x$ is an integer contributes 0, and the remaining $2001 - 14 = 1987$ terms contribute $-1$ each.

Thus, the total sum is:

$$1987 \cdot (-1) + \left\lfloor \frac{4050}{2001} \right\rfloor = -1987 + 2 = -1985.$$

**Final Answer**

$$\boxed{-1985}$$

---

**Example 2: XRPO**

To evaluate the sum

$$\sum_{j=-1000}^{1000} \left\lfloor \frac{2025}{j+0.5} \right\rfloor,$$

we begin by analyzing the structure of the expression. The key transformation is that:

$$j + 0.5 = \frac{2j+1}{2}, \quad \text{so} \quad \frac{2025}{j+0.5} = \frac{4050}{2j+1}.$$

Thus, the sum becomes:

$$\sum_{j=-1000}^{1000} \left\lfloor \frac{4050}{2j+1} \right\rfloor.$$

**Step 1: Splitting the Sum**
We split the sum into three distinctly defined segments:
1. **From $j = -1000$ to $-1$:**
Let $j = -m$, $m = 1$ to 1000, so:

$$\frac{4050}{2j+1} = \frac{4050}{-2m+1} = -\frac{4050}{2m-1}.$$

Then, since $\lfloor x \rfloor = -\left\lceil \frac{4050}{2m-1} \right\rceil$, we get:

$$\left\lfloor \frac{4050}{j+0.5} \right\rfloor = -\left\lceil \frac{4050}{2m-1} \right\rceil.$$

So, the sum becomes:

$$\sum_{m=1}^{1000} -\left\lceil \frac{4050}{2m-1} \right\rceil.$$

2. **At $j = 0$:**

$$\left\lfloor \frac{4050}{1} \right\rfloor = \lfloor 4050 \rfloor = 4050.$$

3. **From $j = 1$ to 1000:** Here, the argument $2j + 1$ is an odd integer in the range from 3 to 2001. Therefore:

$$\left\lfloor \frac{4050}{2j+1} \right\rfloor.$$

**Step 2: Defining the Key Quantities**
Let $F = \sum_{j=1}^{1000} \left\lfloor \frac{4050}{2j+1} \right\rfloor$, $C = \sum_{m=1}^{1000} \left\lceil \frac{4050}{2m-1} \right\rceil$.
Then, the total sum becomes: Total $= 4050 + F - C$.
Now, consider for each odd integer $k = 2m - 1$, from 1 to 1999, the relationship between $\left\lceil \frac{4050}{k} \right\rceil$ and $\left\lfloor \frac{4050}{k} \right\rfloor$.
For each $k$ not dividing 4050, $\left\lceil \frac{4050}{k} \right\rceil > \left\lfloor \frac{4050}{k} \right\rfloor$, so the difference is $-1$.
Thus, the total difference $F - C$ is: $F - C = -N - 4050 + 2$, where $N$ is the number of odd divisors of 4050 in the range from 3 to 1999 (inclusive).

**Step 3: Counting the Odd Divisors of 4050**
The prime factorization of $4050 = 2^1 \cdot 3^4 \cdot 5^2$, so the odd divisors are of the form:

$$3^a \cdot 5^b, \quad a \in \{0, 1, 2, 3, 4\}, \quad b \in \{0, 1, 2\}.$$

Total such divisors: $(4+1)(2+1) = 15$, but we only consider those $\geq 3$ and $\leq 1999$.
After enumerating, we find that **13** of these divisors are between 3 and 1999.
So N = 999 - 13 = 986.
**Step 4: Final Computation**
Now plug into the formula:

$$\text{Total} = 4050 + (F - C) = 4050 + (-N - 4050 + 2) = 2 - N = 2 - 986 = \boxed{-984}.$$

