# OpenReview forum: "XRPO: Pushing the Limits of GRPO with Targeted Exploration and Exploitation"
_ICLR.cc/2026/Conference — Submitted to ICLR 2026_

### Official Review · Reviewer_TrCK · 2025-10-27

**Soundness:** 3
**Presentation:** 3
**Contribution:** 3
**Rating:** 6
**Confidence:** 4

**Summary:**

XRPO (eXplore–eXploit GRPO) is a unified framework designed to **recast policy optimization through the principled lens of rollout exploration–exploitation** within the context of Reinforcement Learning (RL) for Large Language Models (LLMs). Existing methods, such such as GRPO and GSPO, suffer from limited exploration on challenging prompts due to **static rollout allocation** and under-exploitation of informative feedback due to heavy reliance on **sparse rewards**.

XRPO addresses these bottlenecks through three novel mechanisms:

1.  **Hierarchical Rollout Exploration:** A mathematically grounded rollout allocator that **adaptively prioritizes prompts** with higher potential for uncertainty reduction, focusing resources near the decision boundary.
2.  **ICL Seeding:** An in-context seeding strategy that **injects curated exemplars** into zero-reward prompts to break symmetry and steer the model toward more difficult reasoning trajectories.
3.  **Novelty-Guided Advantage Sharpening:** A group-relative mechanism that uses **sequence likelihoods** to amplify correct, low-probability (atypical) responses, extending the policy’s reach beyond simple sparse rewards.

Experiments show that XRPO consistently **outperforms existing advances** (GRPO and GSPO) by up to 4% `pass@1` and 6% `cons@32`, while simultaneously **accelerating training convergence by up to 2.7×**.

**Strengths:**

### Originality

The originality of XRPO stems from its systematic approach to integrate sophisticated exploration and exploitation mechanisms directly into the GRPO framework:

*   **Principled Rollout Allocation:** XRPO introduces a **novel hierarchical rollout planner** that dynamically allocates resources based on two mathematically defined criteria: the expected reduction in statistical uncertainty ($\Delta\hat{q}(n_q)$) and an exploration bonus ($\phi_q(T, n_q)$). This approach is differentiated from existing dynamic sampling methods that simply over-sample or discard prompts.
*   **ICL in RL Framework:** The paper presents **the first demonstration** of using In-Context Learning (ICL) seeding, drawing from an evolving corpus of verified successes, to refine performance on hard prompts with near-zero accuracy within an RL framework. This is a unique method for breaking the "zero-reward symmetry" bottleneck.
*   **Sequence-Level Novelty Measure:** XRPO develops a **sequence-level novelty measure** ($\eta_i = e^{s(y_i)-\bar{s}}$) to guide advantage sharpening, effectively extending the classical entropy bonus from the token level to the full trajectory level. This mechanism allows for fine-grained differentiation among rollouts that receive identical sparse rewards.

### Quality

The work demonstrates high quality through rigorous mechanism design and empirical validation:

*   **Comprehensive Mechanism Integration:** XRPO cohesively integrates three complementary components (Hierarchical Rollout Planning, ICL Seeding, and Advantage Sharpening) to systematically address the limitations of GRPO. Ablation studies confirm that **all three components are necessary** to fully realize the benefits of XRPO, with removal of any module causing consistent performance degradation.
*   **Superior Performance and Efficiency:** XRPO delivers substantial relative improvements, such as **+9.2% in `pass@4` and +20% in `cons@32`** for Qwen2.5-7B-Instruct. Crucially, it achieves **substantially faster training convergence**—up to 2.7× faster on MATH-500 compared to GRPO.
*   **Improved Reasoning Efficiency:** XRPO not only enhances accuracy but also leads to **better inference efficiency**, resulting in substantially shorter response lengths (e.g., a 13.6% reduction on AIME’24). This suggests the model learns to reason in a more targeted and precise fashion.
*   **Demonstrated Precision in Reasoning:** Case studies (Example 1, AIME 2025; Example 2, HMMT 2025) demonstrate XRPO’s superior reasoning rigor and methodological precision compared to GSPO, showing its ability to correctly handle complex boundary conditions and intricate sum splitting.

### Clarity

The paper is well-structured and clearly explains complex concepts:

*   **Detailed Methodology:** The mathematical criteria for rollout prioritization ($\Pi_q$) are clearly defined, and the derivation of the sequence-level novelty measure ($\eta_i$) is explicitly formulated.
*   **Cohesive Algorithm Presentation:** **Algorithm 1** clearly summarizes how the exploration (rollout planning and ICL seeding) and exploitation (advantage sharpening) components are integrated into a cohesive loop.
*   **Reproducibility:** The paper provides a clear reproducibility statement, detailing the models, publicly available datasets, experimental settings, and key hyperparameters ($\lambda_{novelty}=2.5$, $\kappa_{clip}=0.5$).

### Significance

XRPO provides highly significant contributions toward practical and efficient LLM reasoning via RL:

*   **Addressing Fundamental Bottlenecks:** It directly tackles the persistent challenges of **slow training and sparse feedback** inherent in RLVR.
*   **Optimizing Resource Allocation:** The hierarchical rollout planning ensures computational resources are **focused on high-variance prompts near the decision boundary**, where additional rollouts are most informative, thus improving sample efficiency.
*   **Generality and Compatibility:** The exploration–exploitation mechanisms are shown to be **complementary and compatible** with other state-of-the-art methods like GSPO, yielding improved performance when integrated.

**Weaknesses:**

While XRPO offers significant improvements, the following areas could be strengthened with more detailed analysis or discussion of inherent limitations:

#### 1. Limitations Imposed by ICL Seeding Constraints

The effectiveness of ICL seeding, which is critical for breaking zero-reward symmetry, relies on successful retrieval and prompt constraints:

*   **Impact of Truncation on Hard Problems:** The ICL prompt template limits the number of retrieved examples to $K=2$ to conserve context length, and notes that **overly long example solutions are truncated as needed**. For complex, multi-step reasoning questions that consistently fail, truncating the verified successful solution might eliminate critical reasoning steps, potentially reducing the quality of the policy guidance provided by the ICL prompt.
*   **Cold-Start and Retrieval Failure:** The ICL strategy relies on an **evolving corpus** of verified successes. If a prompt is far beyond the model's current capability ("zero-accuracy prompts underexplored," as noted in Section 3.2), there may be no similar solved examples in the corpus. The reliance on similarity search using Qwen3-Embedding-8B means that if retrieval fails, the prompt falls back to its zero-shot form, losing the necessary contextual guidance.

#### 2. Lack of Quantified Runtime Overhead for Dynamic Planning

The hierarchical rollout planning strategy requires periodic re-estimation of prompt statistics ($\bar{r}_q, s_q$) and dynamic allocation across planning rounds.

*   **Unquantified Per-Step Cost:** Although the implementation aims to be "lightweight" and the paper demonstrates overall training *convergence* is accelerated up to 2.7×, the sources **do not explicitly quantify the actual computational overhead** (e.g., latency or GPU time) introduced by the dynamic allocation calculations and the statistical gathering process *per training step*, compared to the fixed-allocation overhead of GRPO/GSPO. A detailed runtime analysis would more fully validate the efficiency claim.

#### 3. Interpretation and Generalization of Novelty-Guided Advantage Sharpening

The novelty mechanism amplifies "low-probability yet correct responses" ($\eta_i < 1$), under the hypothesis that these rare successes are critical learning opportunities that expand the model’s solution repertoire.

*   **Robustness of "Atypical" Solutions:** While the sources show that XRPO learns to produce **shorter, more efficient responses**, it is not deeply analyzed whether these statistically atypical, novelty-boosted trajectories consistently represent logically **more robust, concise, or generally applicable** reasoning paths compared to higher-likelihood correct trajectories. Further analysis is needed to confirm that the novelty measure is reliably capturing high-quality generalization opportunities, rather than merely rewarding successful but statistically rare deviations caused by sampling variance.

**Questions:**

**1. Sensitivity Analysis for the Rollout Planning Trade-off Parameter ($\lambda$):**

*   **Question:** The hierarchical rollout planning priority score $\Pi_q = \Delta\hat{q}(n_q) + \phi_q(T, n_q)$ requires the hyperparameter $\lambda > 0$ to trade off uncertainty-driven exploitation against exploration. Given that $\lambda$ dictates how rollout budgets are distributed across ambiguous vs. sparsely sampled prompts, could the authors provide a **sensitivity analysis** showing how varying the value of $\lambda$ (e.g., $\lambda=0.5, 1.0, 2.0$) affects overall performance metrics (`pass@k`, `cons@32`), similar to the analysis provided for $\lambda_{novelty}$ and $\kappa_{clip}$ in Table 2?

**2. Follow-up Strategy for Persistent Zero-Reward Groups:**

*   **Question:** ICL seeding is activated for prompts where **all rollouts have failed** ("zero-accuracy prompts"). Figures 1b and 4b demonstrate that ICL successfully flips a portion of these hard questions. However, for the persistent group of hard prompts that *still* yield zero successful rollouts even after being augmented by ICL examples in the current training step, how does XRPO manage these prompts in **subsequent training steps**? Are they subjected to repeated ICL seeding, or are they filtered out? Clarification on the long-term management of persistently unsolved groups would enhance understanding of the exploration strategy.

**3. Clarification on Inference Efficiency and Context Management:**

*   **Question:** The paper notes that ICL seeding truncates overly long example solutions to fit the context window. Simultaneously, Figure 4a demonstrates that XRPO achieves better **inference efficiency** by producing shorter average response lengths.
    *   **a) Is there a feedback loop:** Does the preference for shorter (more efficient) successful rollouts, amplified by novelty sharpening, indirectly influence the ICL corpus selection or the successful examples stored, thereby promoting shorter reasoning chains for future ICL seeding?
    *   **b) Impact of Truncation:** Can the authors quantify the fraction of ICL examples that were truncated during training, and whether there was an observable performance difference between ICL-seeded rollouts using truncated vs. untruncated successful examples?

**4. Mathematical Justification for Example 1 Exclusion (Case B):**

*   **Question:** In Example 1 (AIME 2025), XRPO correctly determines $t=10$ by excluding Case B, where $f'(x)=0$ if $\cos(7\pi \sin(5x)) = 0$. This exclusion relies on showing that the condition $\sin(5x) = k/7$ (from $f(x)=0$) cannot simultaneously satisfy $\sin(5x) = (m+1/2)/7$ (from $f'(x)=0$) for integers $k$ and $m$. Could the authors explicitly provide the algebraic proof that this simultaneous satisfaction is impossible, as this strict mathematical filtering is highlighted as key to XRPO's superior accuracy compared to GSPO?

---

> ### Author Response · Authors · 2025-11-22
> **Response to Reviewer TrCK (Part 1)**
>
> Thank you for the thorough and highly encouraging review. We greatly appreciate your recognition of XRPO’s originality, the rigor of its mechanism design, the strength of the empirical results, and its significance for RLVR. We are grateful for your thoughtful evaluation and supportive feedback, and we value the opportunity to address your inquiries, some of which we have reordered for improved clarity and coherence.
>
>
> ***Q1. Impact of Truncation on Hard Problems?***
>
> We agree with the reviewer that truncation of long, multi-step exemplars can, in principle, remove critical reasoning steps and therefore degrade the quality of the ICL seed used during rollouts.
>
> - ***Fraction of Truncated ICL Examples.*** During the construction of the ICL corpus, we include only fully correct sampled solutions. These correct solutions are substantially shorter than failed or partially correct chains of thought. The mean length of a correct example in our corpus is approximately 900 tokens (Table T2). Because each ICL prompt includes only two retrieved exemplars, the combined length of the exemplars plus the query remains comfortably within the allowable context window. As a result, ***we did not observe any truncation of ICL exemplars during training***.
>
> - ***Effect of Truncation on Performance.*** To directly assess the impact of truncation, we performed an ablation in which we artificially truncated long ICL exemplars and compared rollout performance against their full-length versions. As shown in Table T1, truncation leads to a performance drop of 0.7% and 1.7% when both ICL samples are truncated by 5% and 10%, respectively, consistent with the reviewer’s intuition. However, such cases are extremely rare in our actual training setup, where we observe 0% truncation, as shown in Table T2. And even under high truncation rate of 20%, we see that ICL consistently outperforms the baseline
>
> ***Q2. How does XRPO manage all zero prompts in subsequent training steps? Are they subjected to repeated ICL seeding, or are they filtered out?***
>
> XRPO’s behavior on persistently hard prompts is a deliberate design choice and a key differentiator from GRPO-like baselines. GRPO allocates the same number of rollouts to every prompt, which leads to substantial inefficiency because hard prompts repeatedly consume compute while contributing zero learning signal. XRPO explicitly avoids this failure mode: after an initial attempt, prompts that still yield zero reward are down-weighted by the rollout allocator and receive only a small portion of the total rollout budget, with ICL sampling used to recover successes when possible. This allows the remaining budget to be used more judiciously instead of being wasted on repeatedly unsolved cases. These hard prompts then re-enter training naturally in subsequent epochs, where the model has another opportunity to solve them.
>
> ICL seeding is triggered whenever a prompt is identified as being in a zero-accuracy state, but it is not repeatedly applied at every step if no recovery occurs. This avoids unnecessary overhead while preserving the benefits of ICL when it is effective. Overall, XRPO is strictly better than both extremes: GRPO, which wastes compute on unsolved prompts, and selective sampling methods such as DAPO, which ignore them entirely. This is further supported by our empirical results.
>
>
> -----
>
> Table T1: Performance difference between truncated vs. untruncated successful examples
> | Truncation (both samples) | Accuracy |
> |---------------------------|----------|
> | 0%                        | 41.5     |
> | 5%                        | 40.8     |
> | 10%                       | 39.8     |
> | 20%                       | 39.0     |
> | No ICL                    | 37.7     |
>
> Table T2: Training statistics.
>
> |          | ICL Corpus statistics |               | Mean prompt length | Maximum prompt length (with ICL samples). | Maximum prompt length allowed. |
> |----------|-----------------------|---------------|--------------------|-------------------------------------------|--------------------------------|
> |          | Mean                  | 95 percentile |                    |                                           |                                |
> | Qwen3    | 879.9                 | 1,395         | 998                | 5,233                                     | 8,192                          |
> | Qwen2.5  | 711.2                 | 1,219         |                    | 3,826                                     | 4,096                          |

---

> > ### Author Response · Authors · 2025-11-22
> > **Response to Reviewer TrCK (Part 2)**
> >
> > ***Q3. Lack of Quantified Runtime Overhead for Dynamic Planning***
> >
> > Our new results (Table R5) indicate that the per-step latency ratio between XRPO and baseline GRPO is about 1.047, a mere 4.7% overhead.  We use a batch size of 64, 256 rollouts per prompt, and two dynamic rounds to better simulate common training settings. The ICL corpus is loaded once at the start of training (4.22 seconds), and constructing ICL prompts is extremely fast (<0.001 seconds per prompt) due to simple hash-map lookups over precomputed neighbors and responses.
> >
> > Furthermore, we provide the theoretical analysis on XRPO’s latency compared to the baseline in Appendix D.
> >
> > Overall, XRPO not only substantially reduces the number of convergence steps but also maintains a similar per-step latency.
> >
> > ***Q4. Interpretation and Generalization of Novelty-Guided Advantage Sharpening … Preference for shorter (more efficient) successful rollouts***
> >
> > ***Effect of Novelty-Guided Advantage Sharpening.*** We show that the mechanism is in fact free of such bias, based on the length distribution of the shaped entries and the formulation of the novelty score.
> > First, equation (4) in the paper has already shown that each trajectory’s log-likelihood score is normalized by its own length, which removes length bias fundamentally.
> > Second, we computed the response length’s z-score and relative ratio of shaped entries relative to both their full groups and the correct rollouts within those groups(Table R6). The results show that on average, shaped entries’ lengths lie within 0.25 standard deviations and are over 99% close to the mean, indicating no observable length bias.
> >
> > ***XRPO learns to produce shorter, more efficient responses.***  While XRPO does not include an explicit length penalty, the shorter response length is an emergent property of its improved exploration under context constraints. GRPO suffers from inherent "length bias", often generating long and repetitive reasoning chains. On complex problems, these runaway chains hit the model's maximum context limit, resulting in truncation and failure (zero reward). GRPO thus struggles to learn from these long and failed trajectories. XRPO's Rollout Allocator and In-Context Seeding actively guide the model toward valid solution paths for hard prompts. Since a correct solution must be complete within the context window, XRPO effectively filters for trajectories that are concise enough to succeed. This interpretation is further supported by the empirical results shown in Table R8 that ICL reduces average generation length by 34.7% for Qwen3 and 6.17% for Qwen2.5.
> >
> > ***Q5. Sensitivity Analysis for the Rollout Planning Trade-off Parameter ($\lambda$)***
> > We have added new hyperparameter studies for the exploration strength parameter $\lambda$ and the confidence interval parameter $\alpha$ (Table R1), showing that the accuracies fluctuate within a narrow band with a mean of 54.84% and a sample standard deviation of 1.41.
> >
> > ----
> > Table R1: Hyperparameter sensitivity analysis for the exploration strength $\lambda$ and the confidence interval parameter $\alpha$
> >
> > | $\lambda$ | $\alpha$ | Score (%) |
> > |-----------|----------|-----------|
> > | 0.10      | 0.95     | 53.66     |
> > | 0.12      | 0.93     | 53.06     |
> > | 0.08      | 0.95     | 56.27     |
> > | 0.12      | 0.97     | 55.32     |
> > | 0.08      | 0.93     | 55.89     |
> >
> > Table R5: XRPO Latency analysis
> >
> > | Method | Dynamic Planning | Actor Rollout | Construct ICL Prompts | Update ICL Corpus | Advantage Shaping |   Total  |
> > |:------:|:----------------:|:-------------:|:---------------------:|:-----------------:|:-----------------:|:--------:|
> > |  XRPO  |      79.497      |    1448.552   |         6.699         |       11.816      |       0.329       | 1546.893 |
> > |  GRPO  |         -        |    1477.861   |           -           |         -         |         -         | 1477.861 |
> >
> > Table R6: Length distribution analysis for advantage-shaping
> > |     Metric     | w.r.t Correct Rollouts | w.r.t. Full Groups |
> > |:--------------:|:----------------------:|:------------------:|
> > |     Z-score    |          0.257         |       -0.280       |
> > | Relative Ratio |          1.040         |        0.991       |
> >
> > Table R8: ICL reduces average generation length
> > | Response length | Qwen3 | Qwen2.5 |
> > |-----------------|-------|---------|
> > | Without ICL     | 881.7 | 12906.1 |
> > | With ICL        | 827.2 | 8427.9  |
> > | Reduction       | 6.17% | 34.7%   |

---

> > > ### Author Response · Authors · 2025-11-22
> > > **Response to Reviewer TrCK (Part 3)**
> > >
> > > ***Q6. Cold-Start and Retrieval Failure in ICL***
> > >
> > > Our ICL strategy is robust to individual retrieval failures because we do not rely on a single nearest neighbor. Instead of taking only the top-1 retrieved example, XRPO retrieves the top-10 most similar solved prompts, which substantially lowers the chance of failing to provide useful contextual guidance. In rare worst-case scenarios where no sufficiently similar successes exist, the prompt does revert to a zero-shot form, but this simply reflects the current capability frontier of the model. Importantly, because XRPO incorporates generated successes starting from epoch 0, it always maintains a guaranteed lower bound on available exemplars. This corpus naturally grows over training and can be further enhanced with external or domain-specific seed examples, making retrieval failure progressively less likely over time.
> > >
> > > ***Q7. Mathematical Justification for Example 1 Exclusion (Case B)***
> > >
> > > We formally demonstrate that the roots of $f(x)$ and the zeros of the cosine factor in $f'(x)$ are disjoint sets. Let the function be defined as $f(x) = \sin(7\pi \sin(5x))$.
> > >
> > > **Condition for Roots** The function equals zero when its argument is an integer multiple of $\pi$:
> > >
> > > $$7\pi \sin(5x) = k\pi \quad \Longrightarrow \quad \sin(5x) = \frac{k}{7}, \qquad k \in \mathbb{Z}.$$
> > >
> > > **Condition for Case B (Derivative Zero)** The factor $\cos(7\pi \sin(5x))$ vanishes when its argument is an odd multiple of $\frac{\pi}{2}$:
> > >
> > > $$7\pi \sin(5x) = \left(m + \tfrac12\right)\pi \quad \Longrightarrow \quad \sin(5x) = \frac{m + 1/2}{7}, \qquad m \in \mathbb{Z}.$$
> > >
> > >
> > > **Proof That the Two Sets Are Disjoint** If a value of x satisfied both conditions, then
> > > $$\frac{k}{7} = \frac{m + 1/2}{7}$$
> > >
> > > which implies
> > >
> > > $$k = m + \tfrac12.$$
> > >
> > > Since $k$ and $m$ are integers, their difference must also be an integer. The value $\tfrac12$ is not an integer, so this is impossible. Therefore the conditions cannot hold simultaneously.

---

### Official Review · Reviewer_SgHu · 2025-10-28

**Soundness:** 2
**Presentation:** 2
**Contribution:** 2
**Rating:** 2
**Confidence:** 4

**Summary:**

The paper proposes XRPO, a method to improve exploration and exploitation in RL-based reasoning. It combines three components: (1) uncertainty-based sampling budget allocation, (2) ICL seeding for zero-reward prompts, and (3) novelty-aware advantage sharpening that boosts rare but correct trajectories. The goal is to make GRPO training more efficient and effective on reasoning tasks.

**Strengths:**

1. The paper tackles a classic and important problem: the EE balance problem in RLVR. The problem is important and practically relevant for efficient RL training in reasoning-heavy domains.

2. The components address meaningful aspects of this problem: sampling allocation for more informative prompts, ICL seeding to break zero-signal cases, and sharpening advantages for rare but correct responses.

**Weaknesses:**

1. Limited novelty: while each of the three components addresses a meaningful aspect of the problem, similar strategies have been studied in related work. The contribution mostly lies in combining them, which makes the overall novelty limited.

2. Ablation results raise concerns: In Figure 3b, removing any single technique leads to performance close to or below GSPO (pass@1=35.52 / cons@32=45.84 per Table 1 if my understanding is correct). This raises concerns that the gains might not be robust and could come from tuning or noise, rather than stable improvements of each technique.

3. Insufficient experiments: Table 1 only compares XRPO with GSPO on Qwen3-1.7B, while Figure 3a compares GRPO on Qwen2.5-7B-Instruct, but reports different metrics (pass@4 and cons@32) from Table 1 (pass@1 / cons@32). The selective choice of evaluation metrics raises concerns regarding the generality and robustness of the reported improvements. In addition, the settings are not clearly described in the paper, and the reader has to infer them from the context.

4. Unclear training details and insufficient clarity in writing: the paper only briefly mentions using MATH data at Line401, but it’s unclear whether all experiments are trained on MATH or multiple datasets. The data setup needs to be stated explicitly.

5. Inconsistent baselines and metrics: The paper compares XRPO mainly to GSPO and GRPO, with few other strong baselines. There are existing works on prompt selection and external-guidance exploration that should be discussed or compared against.

6. XRPO introduces more complexity compared to GRPO/GSPO, but the benefits are not convincingly demonstrated.

**Questions:**

Please see above weaknesses.

---

> ### Author Response · Authors · 2025-11-22
> **Response to Reviewer SgHu (Part 1)**
>
> We sincerely thank the reviewer for the thoughtful and detailed feedback and appreciate the careful evaluation of our work. We value the opportunity to address your inquiries and, for clarity, have reordered some questions to ensure a more coherent response.
>
> ***Q1: Limited novelty: similar strategies have been studied in related work?***
>
> We respectfully argue that XRPO introduces distinct algorithmic innovations that fundamentally differ from existing approaches. We highlight three key differentiators:
>
> - Unlike dynamic sampling methods such as DAPO, XRPO does not discard zero-accuracy prompts. Instead, it identifies them as exploration targets and actively rescues them by retrieving successful patterns from similar problems.  By proactively leveraging rollout dynamics, this recovers otherwise uninformative training signals into high-value learning opportunities, whereas rejection sampling, priority sampling, and their variants are purely reactive and do not encourage exploration.
> - To the best of our knowledge, XRPO is the first framework to propose ICL seeding in the context of RLHF. We further show that ICL seeding can recover learning signals that meaningfully support training. This allows the model to bootstrap its own improvement in a fully self-contained manner, eliminating the dependency on external teachers or outcome verifiers required by previous methods.
> - Further, a critical limitation of standard RLVR is that successful reasoning paths with low likelihood are downweighted in the reward function, which leads to underexploitation of promising trajectories. XRPO’s Advantage Sharpening addresses this issue, and to the best of our knowledge, we are the first to introduce a correction to the advantage terms in GRPO to mitigate this known RL failure mode.
> - Our ablation studies demonstrate that these components address orthogonal failure modes. ICL Seeding primarily recovers consistency (fixing collapse on hard problems), while Advantage Sharpening drives peak performance (pass@1). The significant performance drop when removing either component proves they are not merely additive, but functionally co-dependent for stable RLVR training.
>
> In Tables R3 and R4, we compared XRPO against DAPO and TreePO. XRPO consistently outperforms these methods across benchmarks, confirming that the specific algorithmic choices in XRPO provide robust gains over the generic strategies they resemble.
>
>
> Table R3: Comparison of GSPO, Dynamic Sampling and TreePO Sampling with XRPO across benchmarks for Qwen3-1.7B (Reasoning).
> |                  |   AIME 	   |    2024   |   AIME    |    2025   |  HMMT  |    2025   |   BRUMO   |    2025   | CodeForces |           |  Average  |
> |------------------|:---------:|:---------:|:---------:|:---------:|:------:|:---------:|:---------:|:---------:|:----------:|:---------:|:---------:|
> |                  |   pass@1  |  const@32 |   pass@1  |  const@32 | pass@1 |  const@32 |   pass@1  |  const@32 |   pass@1   |   pass@1  |  const@32 |
> |       GSPO       |   42.39   | 60.00 |   33.23   |   46.67   |  21.25 |   20.00   |   45.20   |   56.67   |    13.72   |   31.16   |   45.84   |
> | Dynamic Sampling |   40.31   |   43.33   |   32.50   |   33.33   |  20.42 |   23.33   |   39.90   |   43.33   |    9.48    |   28.52   |   35.83   |
> |  TreePO Sampling |   38.33   |   50.00   |   26.04   |   33.33   |  17.29 |   20.00   |   37.40   |   46.67   |    9.51    |   25.71   |   37.50   |
> |       XRPO       | 46.46 |   56.66   | 35.72 | 50.00 |  22.29 | 26.67 | 47.39 | 60.00 |  13.80 | 33.13 | 48.33 |
>
>
> Table R4: Comparison of GRPO, Dynamic Sampling and TreePO Sampling  with XRPO across benchmarks for Qwen2.5-7B
> |                     |        | AIME 2024 |          |        | AIME 2025 |          |        | BRUMO 2025 |          |        | Average |          |
> |---------------------|--------|-----------|----------|:------:|:---------:|:--------:|:------:|:----------:|:--------:|:------:|:-------:|:--------:|
> |                     | pass@1 | pass@4    | const@32 | pass@1 | pass@4    | const@32 | pass@1 | pass@4     | const@32 | pass@1 | pass@4  | const@32 |
> | GRPO                | 10.31  | 18.10     | 13.33    | 6.73   | 15.31     | 10       | 15.83  | 26.30      | 26.67    |  10.96 |  19.90  |   16.67  |
> | Dynamic Sampling    | 10.52  | 20.22     | 16.67    | 6.67   | 15.57     | 10       | 17.92  | 28.72      | 26.67    |  11.70 |  21.50  |   17.78  |
> | TreePO Sampling     | 8.96   | 17.20     | 16.67    | 5.10   | 13.72     | 6.67     | 19.17  | 29.55      | 26.67    |  11.08 |  20.16  |   16.67  |
> | XRPO                | 11.25  | 21.17     | 16.67    | 7.71   | 16.80     | 13.33    | 17.19  | 29.25      | 30.00    |  12.05 |  22.41  |   20.00  |
> | Rel. Gain wrt GRPO. | 9.09%  | 16.96%    | 24.98%   | 14.55% | 9.73%     | 33.30%   | 8.55%  | 11.22%     | 12.53%   |        |         |          |

---

> > ### Author Response · Authors · 2025-11-22
> > **Response to Reviewer SgHu (Part 2)**
> >
> > ***Q2: Ablation results raise concerns: in Figure 3(b), the gains might not be robust and could come from tuning or noise, rather than stable improvements of each technique.***
> >
> > Our goal with Fig. 3(b) is to isolate the contribution of each component of XRPO, not to claim that each ablated variant must individually dominate all baselines.
> > We would like to clarify three points:
> > - ***Full XRPO consistently outperforms very recent state-of-the-art methods, such as GSPO and the newly added DAPO and TreePO.*** As shown in Table 1 and Fig. 3(a) of main paper, the complete XRPO improves over GSPO/GRPO on both Qwen3-1.7B and Qwen2.5-7B-Instruct in pass@k and cons@32 across multiple benchmarks. Although some ablated variants are close to GSPO on certain metrics, the full XRPO is consistently the best-performing method across datasets and metrics.
> > - ***The ablations show structured degradation, not random noise.*** In Fig. 3(b), removing ICL seeding leads to approximately a 4% drop in cons@32, and removing advantage sharpening yields about a 3.5% drop in pass@1. All ablated variants are strictly worse than full XRPO, and different components mainly influence different aspects of performance (ICL primarily improves consistency/cons@32, while advantage sharpening mainly boosts pass@k). This monotonic and structured degradation is more consistent with meaningful contributions of each component than with noise or accidental over-tuning.
> > - ***Additional robustness checks and hyperparameter sensitivity.*** To further address the concern that the gains might come from tuning, we have:
> > 1. Provided sensitivity analyses for the novelty bonus and clamp factor (Table 2 in Main paper);
> > 2. Added new hyperparameter studies for the exploration strength parameter $\lambda$ and the confidence interval parameter  $\alpha$ (Table R1); and
> > 3. Extended our study to another backbone, Llama 3.2 3B  (Table R2), along with additional baselines (Table R3 and Table R4).
> >
> > These results show that XRPO’s improvements are stable over a reasonable range of hyperparameters and are reproducible across different backbones and baselines.
> >
> > ***Q3: Insufficient experiments: Table 1 only compares XRPO with GSPO on Qwen3-1.7B, while Figure 3a compares GRPO on Qwen2.5-7B-Instruct, but reports different metrics (pass@4 and cons@32) from Table 1 (pass@1 / cons@32).***
> >
> > We have revised the experimental section to clarify both the motivation for the metrics and the completeness of reported results.
> >
> > - ***New models and baselines.*** We have expanded our experimental comparisons as follows: (i) We add results comparing XRPO against DAPO [1] and TreePO [2] in Table R3 and Table R4, under the same training and evaluation protocols. (ii) We include additional results on another backbone, Llama 3.2 3B, in Table R2. We observe that XRPO consistently outperforms all baseline methods.
> > - ***Why pass@4 for Qwen2.5-7B-Instruct in Fig. 3(a)?*** Qwen2.5-7B-Instruct is a relatively weaker backbone on the recent, challenging math benchmarks. In this low-accuracy regime, pass@1 often underestimates the relative improvements from RL methods because many prompts remain unsolved at k=1. We therefore reported pass@4 in Fig. 3(a) to better capture gains in the “few-samples” regime while keeping the sampling budget realistic.
> > - ***Added pass@1 results for completeness.*** To address the reviewer’s concern about selective metrics, we have now added pass@1 results for Qwen2.5-7B-Instruct in Table R4. These results confirm that XRPO also improves pass@1 over GRPO on this backbone, in addition to the previously reported gains in pass@4 and cons@32.
> >
> > -----
> >
> > Table R1: Hyperparameter sensitivity analysis for the exploration strength $\lambda$ and the confidence interval parameter $\alpha$
> > | $\lambda$ | $\alpha$ | Score (%) |
> > |-----------|----------|-----------|
> > | 0.10      | 0.95     | 53.66     |
> > | 0.12      | 0.93     | 53.06     |
> > | 0.08      | 0.95     | 56.27     |
> > | 0.12      | 0.97     | 55.32     |
> > | 0.08      | 0.93     | 55.89     |
> >
> > Table R2: Comparison of pass@1 and pass@4 metric on GRPO and XRPO across math benchmarks for Llama 3.2 3B
> >
> > |      | AIME 2024	 |        | AIME 2025 |        | BRUMO 2025 |        | MATH   |        | Average |        |
> > |------|:---------:|:------:|:---------:|:------:|:----------:|:------:|--------|--------|---------|--------|
> > |      |   pass@1  | pass@4 |   pass@1  | pass@4 |   pass@1   | pass@4 | pass@1 | pass@4 |  pass@1 | pass@4 |
> > | GRPO | 7.19      | 16.17  | 0.52      | 1.97   | 4.79       | 7.82   | 43.04  | 62.08  | 13.88   | 22.01  |
> > | XRPO | 9.27      | 18.70  | 0.63      | 2.38   | 3.54       | 9.48   | 43.40  | 61.24  | 14.21   | 22.95  |

---

> > > ### Author Response · Authors · 2025-11-22
> > > **Response to Reviewer SgHu (Part 3)**
> > >
> > > ***Q4: Unclear whether all experiments are trained on MATH or multiple datasets.***
> > >
> > > For all other experiments in the paper, we follow the state-of-the-art work of DAPO to construct the training data by randomly sampling 10K examples from both the DAPO-Math-17k dataset [1] and the DeepCoder Dataset [2], keeping the dataset size practical. For the training convergence experiment (Figure 2 in Main paper), we use Qwen2.5-Math-1.5B with a maximum generation length of 2048 tokens and a maximum prompt length of 1024 tokens. This model is trained entirely on the MATH dataset. We have now added the missing details in our main paper.
> > >
> > > ***Q5: Inconsistent baselines and metrics: The paper compares XRPO mainly to GSPO and GRPO, with few other strong baselines. There are existing works on prompt selection and external-guidance exploration that should be discussed or compared against.***
> > >
> > > We thank the reviewer for the suggestion to broaden the set of baselines and for pointing out related lines of work on prompt selection and external-guidance exploration.
> > > - ***Expanded baselines.*** We have expanded our experimental comparisons as follows:
> > > 1. We add results comparing XRPO against DAPO and TreePO in Table R3, under the same training and evaluation protocols.
> > > 2. We include additional results on another backbone, Llama 3.2 3B, in Table R2. These results show that XRPO continues to provide consistent gains over GRPO/GSPO-style baselines, indicating that the improvements are not specific to a single model architecture.
> > > - ***Why GSPO for Qwen3 and GRPO for Qwen2.5.*** We apologize for the confusion caused by mixing GSPO and GRPO baselines across different models. The choice is driven by how these backbones are typically trained:
> > > 1. Qwen3-1.7B is trained and released with a GSPO-style RLVR pipeline, so we primarily compare XRPO against GSPO on this backbone (Table 1 in Main paper, Table R3).
> > > 2. Qwen2.5-7B-Instruct is commonly used with a GRPO-based training setup, so we report comparisons against GRPO for this model (Fig. 3(a) in Main paper, Table R4).
> > > - ***Positioning XRPO relative to prompt selection and external-guidance methods.***
> > > 1. ***Prompt selection / dynamic sampling (e.g., DAPO).*** We now include DAPO as an explicit baseline (Table R3). In contrast to data-selection methods that may down-weight or discard low-accuracy prompts, XRPO does not discard zero-accuracy prompts. Instead, it turns these “dead” prompts into useful training signals by applying ICL seeding when all rollouts fail, thereby rescuing otherwise uninformative groups without requiring additional labeled data.
> > > 2. ***External-guidance exploration methods.*** Several methods rely on stronger external models or verifiers (e.g., teacher models, powerful reward models, or external solvers) to guide exploration. XRPO is designed for a complementary setting: it does not rely on stronger external models for feedback, and operates purely within the given backbone and reward formulation. We therefore view these methods as related but not directly comparable in terms of resource assumptions; nonetheless, we now discuss them explicitly in Sec. 2 and clarify how XRPO can in principle be combined with such guidance if available.
> > >
> > > ***Q6: XRPO introduces more complexity compared to GRPO/GSPO, but the benefits are not convincingly demonstrated.***
> > >
> > > Our new results (Table R5) indicate that the per-step latency ratio between XRPO and baseline GRPO is about 1.047, mere 4.7% overhead.  We use a batch size of 64, 256 rollouts per prompt, and two dynamic rounds to better simulate common training settings. The ICL corpus is loaded once at the start of training (4.22 seconds), and constructing ICL prompts is extremely fast (<0.001 seconds per prompt) due to simple hash-map lookups over precomputed neighbors and responses.
> > >
> > > Furthermore, we provide the theoretical analysis on XRPO’s latency compared to the baseline in Appendix D in main paper.
> > >
> > > Overall, XRPO not only substantially reduces the number of convergence steps but also maintains a similar per-step latency.
> > >
> > > ------
> > >
> > > Table R5: XRPO Latency analysis
> > >
> > > | Method | Dynamic Planning | Actor Rollout | Construct ICL Prompts | Update ICL Corpus | Advantage Shaping |   Total  |
> > > |:------:|:----------------:|:-------------:|:---------------------:|:-----------------:|:-----------------:|:--------:|
> > > |  XRPO  |      79.497      |    1448.552   |         6.699         |       11.816      |       0.329       | 1546.893 |
> > > |  GRPO  |         -        |    1477.861   |           -           |         -         |         -         | 1477.861 |

---

> ### Author Response · Authors · 2025-11-22
> **Response to Reviewer SgHu (Part 4)**
>
> ***Q7: The novelty metric could bias toward short or lucky outputs. Analyze length sensitivity and show per-length bins or alternate normalization.***
>
> We show that the mechanism is in fact free of such bias, based on the length distribution of the shaped entries and the formulation of the novelty score.
> First, equation (4) in the paper has already shown that each trajectory’s log-likelihood score is normalized by its own length, which removes length bias fundamentally.
> Second, we computed the response length’s z-score and relative ratio of shaped entries relative to both their full groups and the correct rollouts within those groups (Table R6). The results show that on average, shaped entries’ lengths lie within 0.25 standard deviations and are over 99% close to the mean, indicating no observable length bias.
>
> ----
>
> ***References:***
> [1] DAPO, https://arxiv.org/abs/2503.14476
>
> [2] TreePO, https://arxiv.org/abs/2508.17445
>
> [3] DeepCoder, https://www.together.ai/blog/deepcoder
>
> [4] RLEP, https://arxiv.org/html/2507.07451v1
>
> Table R6: Length distribution analysis for advantage-shaping
> |     Metric     | w.r.t Correct Rollouts | w.r.t. Full Groups |
> |:--------------:|:----------------------:|:------------------:|
> |     Z-score    |          0.257         |       -0.280       |
> | Relative Ratio |          1.040         |        0.991       |

---

> > ### Comment · Reviewer_SgHu · 2025-11-26
> >
> > Thank you for the additional experiments and clarifications. Unfortunately, they do not fully address my concern.
> >
> > (1) Current ablations only show that each component is necessary within XRPO; they do not demonstrate that each component's contribution alone to improves over strong baselines. This weakens the claim that each sub-technique is valuable in its own right.
> >
> > (2) For truly hard questions that the model cannot solve, it typically also fails on closely related questions. In this regime, the effectiveness of ICL seeding is unclear. The ICL looks more like a form of data augmentation than a fundamentally new exploration strategy, and its advantage over simply replaying successful trajectories in a standard training buffer is not clearly analyzed. It also remains unclear to what extent ICL seeding actually helps to solve previously unsolved problem types (improving pass@k), or merely improving performance on clusters of similar problems that the model is already capable of solving (reflected in cons@32).
> >
> > (3) The new results in Table 5 show relatively small gains on pass@1 and pass@4 compared to the dynamic sampling baseline. How does XRPO compare to the full DAPO setting? And I am still unsure about the choice of k=4: in many math/code evaluation settings larger k (e.g., 16 or 32) is common.

---

> > > ### Author Response · Authors · 2025-11-27
> > > **Response to Reviewer SgHu (Part 5)**
> > >
> > > We thank the reviewer for the prompt engagement. To address each concern clearly, we break down the question into its key components and respond to each in turn. Please feel free to let us know if any part would benefit from additional clarification. We would be glad to elaborate further.
> > >
> > > ***Q8: Current ablations only show that each component is necessary within XRPO; they do not demonstrate that each component's contribution alone to improves over strong baselines. This weakens the claim that each sub-technique is valuable in its own right.***
> > >
> > > In Table 4 (Appendix B), we provide dataset-wise comparisons for each XRPO component alongside the baselines (TreePO and Dynamic Sampling). As shown in the table, XRPO and all of its ablated variants consistently outperform compared baselines in both averaged pass@1 and cons@32, indicating that each design choice provides meaningful standalone benefits beyond the full method. Notably, even the simplest variant that uses only our Hierarchical Rollout Planning surpasses every baseline, illustrating that our exploration mechanism alone contributes substantial gains.
> > >
> > > XRPO takes an end-to-end approach to solving the explore-exploit problem in GRPO, and each of its components is integral and coherent. For example, attention or MLP layers alone in transformers are not going to outperform the strong baseline because they are complementary to each other. Similarly, we are not claiming any individual components to be sufficient on their own, but note that they already bring measurable benefits individually.

---

> > > > ### Author Response · Authors · 2025-11-27
> > > > **Response to Reviewer SgHu (Part 6)**
> > > >
> > > > ***Q9: For truly hard questions that the model cannot solve, it typically also fails on closely related questions. In this regime, the effectiveness of ICL seeding is unclear.***
> > > >
> > > > We want to kindly note that Figure 1b already shows that ICL seeding achieves a 15–20% flip rate among zero-reward prompts, which demonstrates that it can meaningfully convert a nontrivial portion of hard failures into successful rollouts. So the effectiveness of ICL seeding in this regime is clear. Incorporating this mechanism is therefore strictly better than omitting it, even in the challenging regime the reviewer highlights.
> > > >
> > > > At the same time, XRPO does not rely exclusively on ICL seeding. We agree that, for some problems, the current policy may simply be incapable of producing a correct solution regardless of the provided examples. XRPO is designed to handle this gracefully: when the seeded rollout still fails, the method skips that branch rather than spending substantial compute on unproductive rollouts. This selective exploration contributes to XRPO’s fast convergence.
> > > >
> > > > Finally, XRPO operates entirely within its own backbone and reward formulation, without relying on stronger external teacher models or verifiers. While this places a natural limit on how much ICL alone can assist with the hardest problems, it still provides reliable gains within this constrained setting.
> > > >
> > > > ***Q10: The ICL looks more like a form of data augmentation than a fundamentally new exploration strategy, its advantage over simply replaying successful trajectories in a standard training buffer is not clearly analyzed.***
> > > >
> > > > Replay buffers [4] in RLHF typically store successful trajectories for the same prompt, so they cannot help on instances where no success has ever been observed. This is exactly the zero-reward degeneracy described in Section 4.1. In contrast, XRPO’s ICL seeding draws verified solutions from similar questions, injecting informative structure into otherwise unsolved prompts and breaking this symmetry. In this narrow sense, replay can be viewed as a lower-bound case of ICL, which generalizes beyond a single prompt.
> > > >
> > > > Moreover, ICL seeding does more than augment past rollouts. By transferring knowledge across related problems, it creates stable reasoning patterns that improve consistency across samples, as reflected in higher cons@32 scores. This cross-prompt benefit cannot be achieved by standard replay.
> > > >
> > > > Finally, ICL helps address a major limitation of GRPO, where long and repetitive reasoning chains often exceed the context window and yield zero reward. By guiding generation toward trajectories that remain within the window, XRPO produces more efficient and concise solutions. This is reflected in Table 7 (Appendix F), which shows a 34.7 percent reduction in average length for Qwen3 and a 6.17 percent reduction for Qwen2.5.
> > > >
> > > > ***Q11: It also remains unclear to what extent ICL seeding actually helps to solve previously unsolved problem types (improving pass@k), or merely improving performance on clusters of similar problems that the model is already capable of solving (reflected in cons@32)***
> > > >
> > > > We provide empirical evidence that ICL seeding contributes to both effects. First, Figure 1b shows that ICL seeding achieves a 15-20 percent flip rate among zero-reward prompts, indicating that it offers effective external guidance on problems the model previously could not solve. Consistently, Figure 1a shows a 4-6 percent absolute improvement when ICL is enabled.
> > > >
> > > > Second, Table 4 (Appendix B) shows that removing ICL alone leads to a substantial drop in cons@32, underscoring its importance for consistency. Together, these results show that ICL seeding both helps unlock previously unsolved instances and strengthens behavior on clusters of similar tasks.

---

> > > > > ### Author Response · Authors · 2025-11-27
> > > > > **Response to Reviewer SgHu (Part 7)**
> > > > >
> > > > > ***Q12:  The new results in Table 5 show relatively small gains on pass@1 and pass@4 compared to the dynamic sampling baseline.***
> > > > >
> > > > > We consistently match or surpass Dynamic Sampling across all datasets in Table 5, and XRPO provides an additional 3 percent gain in cons@32, indicating more stable performance. Notably, for the more capable Qwen3 model, XRPO achieves a 4.5% and 11.67% improvement in pass@1 and pass@32, respectively, over Dynamic Sampling, further highlighting its advantage.
> > > > >
> > > > > ***Q13: How does XRPO compare to the full DAPO setting?***
> > > > >
> > > > > In our responses, we use DS to denote the DAPO setup, and ensure a fair comparison with XRPO.
> > > > > Aside from that, we follow the same setup as DAPO, including token-level loss and Clip-Higher, and we apply these settings uniformly to the GRPO and GSPO baselines as well, following recent state-of-the-art setups.
> > > > >
> > > > > ***Q14: I am still unsure about the choice of k=4: in many math/code evaluation settings larger k (e.g., 16 or 32) is common.***
> > > > >
> > > > > Our original choice of k=4 was not intended to carry any special significance. We selected a small k to avoid artificially inflating performance whisle still reflecting the model’s underlying capability. Following reviewers' suggestion, we have now included pass@1 in Table 5, and for completeness, we also report pass@16 and pass@32 in Table S1. These new experiments demonstrate that XRPO consistently outperforms all baselines, even under larger values of k.
> > > > >
> > > > >
> > > > > Table S1: Comparison of Qwen2.5-7B on pass@16 and pass@32
> > > > >
> > > > > | Method            | Metric   | AIME'24 | AIME'25 | BRUMO'25 | Avg.  |
> > > > > |:------------------|:---------|:--------|:--------|:----------|:-------|
> > > > > | GRPO              | pass@16  | 22.79   | 26.05   | 33.92     | 27.59 |
> > > > > |                   | pass@32  | 26.67   | 30.00   | 36.67     | 31.11 |
> > > > > | Dynamic Sampling  | pass@16  | 32.53   | 25.79   | 39.24     | 32.52 |
> > > > > |                   | pass@32  | 40.00   | 30.00   | 46.67     | 38.89 |
> > > > > | TreePO Sampling   | pass@16  | 22.45   | 27.40   | 41.53     | 30.46 |
> > > > > |                   | pass@32  | 23.33   | 36.67   | 46.67     | 35.56 |
> > > > > | XRPO              | pass@16  | 30.82   | 29.04   | 42.41     | 34.09 |
> > > > > |                   | pass@32  | 40.00   | 36.67   | 50.00     | 42.22 |

---

### Official Review · Reviewer_Krpv · 2025-10-31

**Soundness:** 3
**Presentation:** 3
**Contribution:** 3
**Rating:** 6
**Confidence:** 4

**Summary:**

This paper proposes XRPO to improve RLVR-style training for reasoning LLMs by explicitly balancing exploration (adaptive rollout allocation & ICL seeding for zero-reward prompts) and exploitation (novelty-guided advantage sharpening).
This paper through uncertainty-driven hierarchical allocation + UCB-style exploration, finite rollouts are intelligently allocated to prompts that maximize reduction of statistical error; when encountering completely incorrect groups, ICL seeding (small-sample prompts) breaks the zero variance dilemma under 0/1 rewards;
For sequences already judged correct, this paper employ advantage sharpening based on sequence novelty to expand the policy boundary. The experiments demonstrate higher accuracy and faster convergence across multiple math and coding benchmarks, alongside shorter and more decisive inference lengths.

**Strengths:**

This paper tackles a clear weakness of GRPO and does so with a simple yet principled exploration–exploitation design. The rollout allocator and novelty-based advantage adjustment both make intuitive sense.

The results are strong across both math and coding tasks, with faster convergence and better reasoning efficiency.

The ablation and sensitivity studies are thorough enough to show that each part of XRPO matters.

The way of how ICL seeding is used to 'revive' dead prompts is insightful — it’s a practical fix for zero-reward cases that actually shows measurable impact.

**Weaknesses:**

I think the baseline comparisons are a bit narrow, mostly GRPO and GSPO. I’d like to see results against other dynamic rollout or sampling-based RL methods for a fairer picture.

The reported speedup is in training steps, not wall-clock time. It’s unclear how much overhead the rollout allocator and ICL retrieval add in real runtime and I don't find more informations about this profilling work.

The novelty metric could bias toward short or lucky outputs. Some length-controlled analysis or ablations would help clarify that. I strongly suggest the authors analyze length sensitivity and show per-length bins or alternate normalization.

The priority $ \hat{\Delta}_q $ relies on sample std $ s_q $ and t-critical values; when $ s_q \to 0 $ early, the exploration bonus $ \phi_q $ rescues, but practical stability under heavy class imbalance or reward mis-calibration is not deeply analyzed; a toy-simulation study could support the allocator’s robustness.

The ICL corpus reuse raises my mild concerns about data leakage, more explicit filtering or similarity checks would make the claim cleaner.

All experiments in this paper are on Qwen series models; broader model coverage (e.g., Open sourced models like Llama, deepseek series) would strengthen generality.

**Questions:**

1. Can you share wall-clock or GPU-hour comparisons or detailed profill results to confirm the claimed speedup? Including overhead from allocation, novelty scoring, and ICL retrieval, to substantiate the 2.4–2.7× speedup.
2. How do you deduplicate ICL examples against eval? Any semantic similarity thresholds to prevent near-leakage?
3. Could you add controlled comparisons to DAPO / selective rollouts / tree-search RL under the same total rollout budget and same decoding?
4. Have you ever done the fine-gained ablation studies on per-dataset ablations instead of the final average multiple datasets result?

---

> ### Author Response · Authors · 2025-11-22
> **Response to Reviewer Krpv (Part 1)**
>
> Thank you for the positive and thoughtful assessment of our work. We appreciate your recognition of XRPO’s principled exploration–exploitation design, the strength and consistency of the results across domains, and the value of our ablation and sensitivity studies. We value the opportunity to address your inquiries and, for clarity, have reordered some questions to ensure a more coherent response.
>
>
> ***Q1: Could you add controlled comparisons to DAPO / selective rollouts / tree-search RL under the same total rollout budget and same decoding? Also, broader model coverage (e.g., Open sourced models like Llama, deepseek series).***
>
> We have expanded our experimental comparisons as follows:
> - We add results comparing XRPO against DAPO and TreePO in Table R3 and R4, under the same training and evaluation protocols.
> - We include additional results on another backbone, Llama 3.2 3B, in Table R2.
>
> These results show that XRPO continues to provide consistent gains over GRPO-style baselines, indicating that the improvements are not specific to a single model architecture.
>
> -----
>
> Table R2: Comparison of pass@1 and pass@4 metric on GRPO and XRPO across math benchmarks for Llama 3.2 3B
>
> |      | AIME 2024	 |        | AIME 2025 |        | BRUMO 2025 |        | MATH   |        | Average |        |
> |------|:---------:|:------:|:---------:|:------:|:----------:|:------:|--------|--------|---------|--------|
> |      |   pass@1  | pass@4 |   pass@1  | pass@4 |   pass@1   | pass@4 | pass@1 | pass@4 |  pass@1 | pass@4 |
> | GRPO | 7.19      | 16.17  | 0.52      | 1.97   | 4.79       | 7.82   | 43.04  | 62.08  | 13.88   | 22.01  |
> | XRPO | 9.27      | 18.70  | 0.63      | 2.38   | 3.54       | 9.48   | 43.40  | 61.24  | 14.21   | 22.95  |
>
>
> Table R3: Comparison of GSPO, Dynamic Sampling and TreePO Sampling with XRPO across benchmarks for Qwen3-1.7B (Reasoning).
> |                  |   AIME 	   |    2024   |   AIME    |    2025   |  HMMT  |    2025   |   BRUMO   |    2025   | CodeForces |           |  Average  |
> |------------------|:---------:|:---------:|:---------:|:---------:|:------:|:---------:|:---------:|:---------:|:----------:|:---------:|:---------:|
> |                  |   pass@1  |  const@32 |   pass@1  |  const@32 | pass@1 |  const@32 |   pass@1  |  const@32 |   pass@1   |   pass@1  |  const@32 |
> |       GSPO       |   42.39   | 60.00 |   33.23   |   46.67   |  21.25 |   20.00   |   45.20   |   56.67   |    13.72   |   31.16   |   45.84   |
> | Dynamic Sampling |   40.31   |   43.33   |   32.50   |   33.33   |  20.42 |   23.33   |   39.90   |   43.33   |    9.48    |   28.52   |   35.83   |
> |  TreePO Sampling |   38.33   |   50.00   |   26.04   |   33.33   |  17.29 |   20.00   |   37.40   |   46.67   |    9.51    |   25.71   |   37.50   |
> |       XRPO       | 46.46 |   56.66   | 35.72 | 50.00 |  22.29 | 26.67 | 47.39 | 60.00 |  13.80 | 33.13 | 48.33 |
>
>
> Table R4: Comparison of GRPO, Dynamic Sampling and TreePO Sampling  with XRPO across benchmarks for Qwen2.5-7B
> |                     |        | AIME 2024 |          |        | AIME 2025 |          |        | BRUMO 2025 |          |        | Average |          |
> |---------------------|--------|-----------|----------|:------:|:---------:|:--------:|:------:|:----------:|:--------:|:------:|:-------:|:--------:|
> |                     | pass@1 | pass@4    | const@32 | pass@1 | pass@4    | const@32 | pass@1 | pass@4     | const@32 | pass@1 | pass@4  | const@32 |
> | GRPO                | 10.31  | 18.10     | 13.33    | 6.73   | 15.31     | 10       | 15.83  | 26.30      | 26.67    |  10.96 |  19.90  |   16.67  |
> | Dynamic Sampling    | 10.52  | 20.22     | 16.67    | 6.67   | 15.57     | 10       | 17.92  | 28.72      | 26.67    |  11.70 |  21.50  |   17.78  |
> | TreePO Sampling     | 8.96   | 17.20     | 16.67    | 5.10   | 13.72     | 6.67     | 19.17  | 29.55      | 26.67    |  11.08 |  20.16  |   16.67  |
> | XRPO                | 11.25  | 21.17     | 16.67    | 7.71   | 16.80     | 13.33    | 17.19  | 29.25      | 30.00    |  12.05 |  22.41  |   20.00  |
> | Rel. Gain wrt GRPO. | 9.09%  | 16.96%    | 24.98%   | 14.55% | 9.73%     | 33.30%   | 8.55%  | 11.22%     | 12.53%   |        |         |          |

---

> > ### Author Response · Authors · 2025-11-22
> > **Response to Reviewer Krpv (Part 2)**
> >
> > ***Q2: How much overhead the rollout allocator and ICL retrieval add in real runtime.***
> >
> > Our new results (Table R5) indicate that the per-step latency ratio between XRPO and baseline GRPO is about 1.047, mere 4.7% overhead.  We use a batch size of 64, 256 rollouts per prompt, and two dynamic rounds to better simulate common training settings. The ICL corpus is loaded once at the start of training (4.22 seconds), and constructing ICL prompts is extremely fast (<0.001 seconds per prompt) due to simple hash-map lookups over precomputed neighbors and responses.
> >
> > Furthermore, we provide the theoretical analysis on XRPO’s latency compared to the baseline in Appendix D in main paper..
> >
> > Overall, XRPO not only substantially reduces the number of convergence steps but also maintains a similar per-step latency.
> >
> > ***Q3: The ICL corpus reuse raises my mild concerns about data leakage. How do you deduplicate ICL examples against eval? Any semantic similarity thresholds to prevent near-leakage?***
> >
> > Both XRPO and all baselines are trained on the same public DAPO training corpus and DeepCoder datasets, and the ICL corpus is constructed entirely from these same sources. As a result, all methods operate under identical conditions: identical training data, rollout budgets, and reward setups. XRPO differs only in how it allocates rollouts using that corpus. Consequently, even if there were any residual overlap between the training set and the evaluation benchmarks, such overlap would affect all methods equally. Moreover, during evaluation, the ICL corpus is not used at all; all methods, including XRPO, are evaluated with standard prompting, without any retrieval or in-context augmentation.
> >
> > ***Q4: Have you ever done the fine-gained ablation studies on per-dataset ablations?***
> >
> > We provide a detailed breakdown per dataset in Table R7.
> >
> >
> > ***Q5: Analyze length sensitivity and show per-length bins or alternate normalization?***
> >
> > We show that the mechanism is in fact free of such bias, based on the length distribution of the shaped entries and the formulation of the novelty score.
> >
> > First, equation (4) in the paper has already shown that each trajectory’s log-likelihood score is normalized by its own length, which removes length bias fundamentally.
> >
> > Second, we computed the response length’s z-score and relative ratio of shaped entries relative to both their full groups and the correct rollouts within those groups (Table R6). The results show that on average, shaped entries’ lengths lie within 0.25 standard deviations and are over 99% close to the mean, indicating no observable length bias.
> >
> > ----
> >
> > Table R5: XRPO Latency analysis
> > | Method | Dynamic Planning | Actor Rollout | Construct ICL Prompts | Update ICL Corpus | Advantage Shaping |   Total  |
> > |:------:|:----------------:|:-------------:|:---------------------:|:-----------------:|:-----------------:|:--------:|
> > |  XRPO  |      79.497      |    1448.552   |         6.699         |       11.816      |       0.329       | 1546.893 |
> > |  GRPO  |         -        |    1477.861   |           -           |         -         |         -         | 1477.861 |
> >
> > Table R6: Length distribution analysis for advantage-shaping
> > |     Metric     | w.r.t Correct Rollouts | w.r.t. Full Groups |
> > |:--------------:|:----------------------:|:------------------:|
> > |     Z-score    |          0.257         |       -0.280       |
> > | Relative Ratio |          1.040         |        0.991       |
> >
> >
> > Table R7: Dataset wise comparison of XRPO components and baselines.
> >
> > |                  | AIME 2024 |          | AIME 2025 |          | HMMT 2025 |          | BRUMO 2025 |          | Average |         |
> > |------------------|:---------:|:--------:|:---------:|:--------:|:---------:|:--------:|:----------:|:--------:|:-------:|:-------:|
> > |                  | pass@1    | const@32 | pass@1    | const@32 | pass@1    | const@32 | pass@1     | const@32 | pass@1  | cons@32 |
> > | XRPO             | 46.46     | 56.66    | 35.72     | 50.00    | 21.66     | 23.33    | 47.39      | 60.00    | 37.81   | 47.50   |
> > | XRPO w/o AS      | 39.17     | 53.33    | 33.13     | 40.00    | 20.10     | 26.67    | 45.00      | 60.00    | 34.35   | 45.00   |
> > | XRPO w/o ICL     | 44.58     | 53.33    | 35.94     | 43.33    | 20.94     | 20.00    | 46.04      | 56.67    | 36.87   | 43.33   |
> > | XRPO w/o AS+ICL  | 40.72     | 46.66    | 31.45     | 40.00    | 19.06     | 20.00    | 46.04      | 60.00    | 34.32   | 41.67   |
> > | Dynamic Sampling | 40.31     | 43.33    | 32.50     | 33.33    | 20.42     | 23.33    | 39.90      | 43.33    | 33.28   | 35.83   |
> > | TreePO Sampling  | 38.33     | 50.00    | 26.04     | 33.33    | 17.29     | 20.00    | 37.40      | 46.67    | 29.77   | 37.5    |

---

> > > ### Author Response · Authors · 2025-11-22
> > > **Response to Reviewer Krpv (Part 3)**
> > >
> > > ***Q6: The priority $\hat{\Delta}_q$ relies on sample std $s_q$ and t-critical values; when $s_q \to 0$ early, the exploration bonus $\phi_q$ rescues, but practical stability under heavy class imbalance or reward mis-calibration is not deeply analyzed; a toy-simulation study could support the allocator’s robustness.***
> > >
> > > To address the concern regarding stability when $s_q \to 0$ (e.g., due to class imbalance or reward mis-calibration), we conducted a toy-simulation study. We constructed a batch of 8 prompts covering the full variance range, specifically including collapsed prompts where initial samples yielded zero variance.
> > >
> > > ***Setup:*** We allocated 80 base rollouts (10 per prompt) and a dynamic budget of 80 across two rounds. The prompts were categorized by empirical variance: High ($\sim 0.25$), Mid ($0.16$), Low ($0.09$), and Collapsed ($0.00$).
> > >
> > > Results & Analysis (Table K1):
> > > - ***Round 1 (Variance-Driven):*** As expected, the allocation is initially uneven ($Var_{alloc} = 5.50$). The priority $\hat{\Delta}_q$ dominates, causing collapsed prompts ($p_2, p_3$) to receive the minimum allocation (2), while High-Variance prompts receive the maximum (7).
> > > - ***Round 2 (The "Rescue"):*** The exploration bonus $\phi_q$ successfully rescues the starved prompts. Because $n_q$ remained low for the Collapsed group in Round 1, their exploration bonus increases. Consequently, in Round 1, the Collapsed prompts receive the highest allocation (6) surpassing even the High-Variance prompts (5).
> > >
> > > ***System Stability.*** The Allocation Variance row confirms convergence. The variance between prompt allocations drops drastically from 5.50 to 0.25.
> > >
> > > ***Conclusion:*** This confirms that XRPO is self-balancing. Even when $s_q \to 0$ early, the exploration term $\phi_q$ prevents starvation and drives the system toward a stable equilibrium where the total allocation (final row) is equitable across all difficulty levels.
> > >
> > > ----
> > >
> > > Table K1:  Dynamic allocation example
> > > |                 |  P0  |  P1  | P2 | P3 |  P4  |  P5  |  P6  |  P7  | Alloc. Variance |
> > > |-----------------|:----:|:----:|:--:|:--:|:----:|:----:|:----:|:----:|:---------------:|
> > > |       Var       | 0.25 | 0.21 |  0 |  0 | 0.25 | 0.16 | 0.09 | 0.09 |                 |
> > > |    Base Round   |  10  |  10  | 10 | 10 |  10  |  10  |  10  |  10  |        -        |
> > > | Dynamic Round 1 |   7  |   7  |  2 |  2 |   7  |   7  |   2  |   6  |       5.50      |
> > > | Dynamic Round 2 |   5  |   5  |  6 |  5 |   5  |   4  |   5  |   5  |       0.25      |
> > > |      Total      |  22  |  22  | 18 | 17 |  22  |  21  |  17  |  21  |        -        |

---

### Official Review · Reviewer_zmm5 · 2025-11-01

**Soundness:** 2
**Presentation:** 2
**Contribution:** 2
**Rating:** 4
**Confidence:** 4

**Summary:**

The paper proposes XRPO (eXplore–eXploit GRPO), a reinforcement learning method designed to improve the reasoning ability of LLMs. XRPO extends Group Relative Policy Optimization (GRPO) by introducing three main operations: (1) Hierarchical Rollout Planning, which allocates rollouts based on statistical uncertainty and exploration bonuses; (2) In-Context Learning Seeding, which helps overcome zero-reward prompts by providing relevent solved examples; and (3) Novelty-Guided Advantage Sharpening, which amplifies rare but correct responses to enhance exploitation. Experiments across math reasoning and code generation benchmarks demonstrate improvements over GRPO and GSPO.

**Strengths:**

- The paper targets two fundamental limitations of existing RLVR methods—under-exploration and under-exploitation.
- The experimental evaluation is comprehensive, covering diverse benchmarks across reasoning and code generation tasks.

**Weaknesses:**

- The method appears to be a combination of specific techniques without substantial novelty, rather than a broadly generalizable exploration–exploitation framework. The three designs presented in the paper are one of many means to achieve exploration (design 1,2) and exploitation (design 3), and naming it XRPO (eXplore–eXploit GRPO) seems somewhat overstated, as many methods already incorporate or implicitly address exploration and exploitation. Similarly, describing it in Line 17 as "a unified framework that recasts policy optimization through the principled lens of rollout exploration–exploitation" feels like an overclaim.
- The method's practicality for deployment is hindered by its many hyperparameters and specific design choices, such as $\lambda$, confidence $\alpha$, auxiliary model-dependent ICL seeding, and Eq. (6) with  $\lambda_{novelty}$, $\kappa_{clip}$.
- A key hypothesis behind the Novelty-Guided Advantage Sharpening is that "correct solutions with relatively low likelihood compared to other rollouts can drive the most effective learning in LLM reasoning" (Line 173). However, this claim is not sufficiently supported in the paper.
- The BACKGROUND AND MOTIVATION section (Section 3.2) already includes parts of the method and results, which is structurally inappropriate.
- The final priority score for allocating rollouts is defined based on the sample mean and standard deviation of rewards for each prompt. This requires re-estimation in each phase and fails to function in the first phase, despite there being only three phases in total.
- The ICL seeding corpus evolves during training, but its scalability and retrieval efficiency are not adequately analyzed.
- Typo: Figure 4(c) — "Incomplete Response."

**Questions:**

- Figure 4(a): Why does XRPO produce shorter response lengths than GRPO? The paper introduces no explicit mechanism that reduces generation length.
- How does the computational overhead of dynamic rollout allocation compare to static allocation?

---

> ### Author Response · Authors · 2025-11-22
> **Response to Reviewer zmm5 (Part 1)**
>
> Thank you for recognizing our focus on addressing under-exploration and under-exploitation in RLVR, as well as for noting the breadth of our experimental evaluation. We appreciate your positive assessment and the time you took to review our work. We value the opportunity to address your inquiries and, for clarity, have reordered some questions to ensure a more coherent response.
>
> ***Q1: Without substantial novelty? Describing it as "a unified framework that recasts policy optimization through the principled lens of rollout exploration–exploitation" feels like an overclaim.***
>
> We appreciate the reviewer’s perspective on the framing of our method. While we acknowledge that exploration and exploitation are fundamental concepts, our claim of a 'unified framework' stems from the interdependent nature of our three integrated mechanisms, which are designed to address coupled failure modes in GRPO.
>
> - Unlike dynamic sampling methods such as DAPO, XRPO does not discard zero-accuracy prompts. Instead, it identifies them as exploration targets and actively rescues them by retrieving successful patterns from similar problems.  By proactively leveraging rollout dynamics, this recovers otherwise uninformative training signals into high-value learning opportunities, whereas rejection sampling, priority sampling, and their variants are purely reactive and do not encourage exploration.
> - To the best of our knowledge, XRPO is the first framework to propose ICL seeding in the context of RLHF. We further show that ICL seeding can recover learning signals that meaningfully support training. This allows the model to bootstrap its own improvement in a fully self-contained manner, eliminating the dependency on external teachers or outcome verifiers required by previous methods.
> - Further, a critical limitation of standard RLVR is that successful reasoning paths with low likelihood are downweighted in the reward function, which leads to under exploitation of promising trajectories. XRPO’s Advantage Sharpening addresses this issue, and to the best of our knowledge, we are the first to introduce a correction to the advantage terms in GRPO to mitigate this known RL failure mode.
> - Our ablation studies demonstrate that these components address orthogonal failure modes. ICL Seeding primarily recovers consistency (fixing collapse on hard problems), while Advantage Sharpening drives peak performance (pass@1). The significant performance drop when removing either component proves they are not merely additive, but functionally co-dependent for stable RLVR training.
>
> We describe this framework as 'principled' because it strictly adheres to the constraints of self-contained RL, avoiding reliance on external teachers or stronger reward models. We thank you for the feedback on the paper's positioning and will refine the wording in the final version.
>
> ***Q2: Many hyperparameters and specific design choices?***
>
> We appreciate the concern about practicality and hyperparameter burden. XRPO does introduce a small number of additional knobs on top of standard GRPO/GSPO, but we:
> - ***Provide targeted ablations and robustness analyses.*** We added hyperparameter sensitivity studies:
> 1. For the novelty bonus and clamp factor (Table 2 in Main paper);
> 2. Added new hyperparameter studies for the exploration strength parameter $\lambda$ and the confidence interval parameter  $\alpha$ (Table R1);
> - ***Default settings transfer across (new) models.*** We use the same set of XRPO hyperparameters across Qwen3-1.7B (Table R3), Qwen2.5-7B-Instruct (Table R4), and the newly added study on Llama 3.2 3B (Table R2) without per-model re-tuning. This indicates that practitioners can adopt XRPO with a single default configuration.
>
> -----
>
> Table R1: Hyperparameter sensitivity analysis for the exploration strength $\lambda$ and the confidence interval parameter $\alpha$
>
> | $\lambda$ | $\alpha$ | Score (%) |
> |-----------|----------|-----------|
> | 0.10      | 0.95     | 53.66     |
> | 0.12      | 0.93     | 53.06     |
> | 0.08      | 0.95     | 56.27     |
> | 0.12      | 0.97     | 55.32     |
> | 0.08      | 0.93     | 55.89     |
>
> Table R2: Comparison of pass@1 and pass@4 metric on GRPO and XRPO across math benchmarks for Llama 3.2 3B
>
> |      | AIME 2024	 |        | AIME 2025 |        | BRUMO 2025 |        | MATH   |        | Average |        |
> |------|:---------:|:------:|:---------:|:------:|:----------:|:------:|--------|--------|---------|--------|
> |      |   pass@1  | pass@4 |   pass@1  | pass@4 |   pass@1   | pass@4 | pass@1 | pass@4 |  pass@1 | pass@4 |
> | GRPO | 7.19      | 16.17  | 0.52      | 1.97   | 4.79       | 7.82   | 43.04  | 62.08  | 13.88   | 22.01  |
> | XRPO | 9.27      | 18.70  | 0.63      | 2.38   | 3.54       | 9.48   | 43.40  | 61.24  | 14.21   | 22.95  |

---

> > ### Author Response · Authors · 2025-11-22
> > **Response to Reviewer zmm5 (Part 2)**
> >
> > ***Q3: A key hypothesis behind the Novelty-Guided Advantage Sharpening is that "correct solutions with relatively low likelihood compared to other rollouts can drive the most effective learning in LLM reasoning" (Line 173). However, this claim is not sufficiently supported in the paper.***
> >
> > We thank the reviewer for highlighting the need to better justify this hypothesis. Our method relies on a simple but important restriction: novelty-guided advantage sharpening is applied only among correct rollouts, i.e., among trajectories that receive a positive reward under the outcome reward model (ORM).
> > - ***What we can and cannot observe.*** Since the ORM provides a binary correctness signal, we do not attempt to infer which incorrect rollouts are “almost correct” the ORM does not support such fine-grained credit assignment. Instead, our mechanism (1) identifies the subset of correct rollouts for a prompt; (2) upweights those with relatively lower model likelihood (novel/rare correct solutions) and downweights overused high-likelihood ones. Moreover, shaping advantages for incorrect rollouts is risky because these relatively low likelihood generations may be contaminated with repetition or multilingual noise. Such corrupted signals can confuse the model or even poison training.
> > - ***Why is this safe and useful.*** Within the set of correct rollouts, emphasizing lower-likelihood ones cannot decrease the expected correctness reward for that prompt: we are only reweighting different positive trajectories. Intuitively, a greater diversity of correct reasoning paths increases the chance that the model generalizes to structurally similar but unseen problems.
> > - ***Advantage sharpening helps improve pass@k.*** As shown in Table R7, removing advantage sharpening yields a consistent ~3.5-point drop in pass@1, indicating that it provides a stable and meaningful contribution rather than adding noise.
> >
> > ***Q4: The BACKGROUND AND MOTIVATION section (Section 3.2) already includes parts of the method and results, which is structurally inappropriate.***
> >
> > We thank you for the feedback on paper writing. We are happy to fix it in our final version.
> >
> > ***Q5: The final priority score for allocating rollouts is defined based on the sample mean and standard deviation of rewards for each prompt. This requires re-estimation in each phase and fails to function in the first phase, despite there being only three phases in total.***
> >
> > This is by design, and we clarify this more explicitly in the revised text.
> > - ***Cold-start handling.*** In the first phase, we allocate rollouts uniformly across prompts, precisely to obtain an initial estimate of the mean and variance of rewards. This addresses the cold-start issue the reviewer points out: the priority score is intentionally not used before we have sufficient statistics.
> > - ***Subsequent phases.*** Starting from the second phase, we use the empirical mean and standard deviation to compute the priority scores (Eq. (4)/(5)) and dynamically reallocate rollouts.
> >
> > ***Q6: Typo: Figure 4(c) — "Incomplete Response."***
> >
> > Thank you for catching this. We will correct the typo in Figure 4(c) (“Incomplete Response”) in the revised manuscript.
> >
> > -----
> >
> > Table R7: Dataset wise comparison of XRPO components and baselines.
> >
> > |                  | AIME 2024 |          | AIME 2025 |          | HMMT 2025 |          | BRUMO 2025 |          | Average |         |
> > |------------------|:---------:|:--------:|:---------:|:--------:|:---------:|:--------:|:----------:|:--------:|:-------:|:-------:|
> > |                  | pass@1    | const@32 | pass@1    | const@32 | pass@1    | const@32 | pass@1     | const@32 | pass@1  | cons@32 |
> > | XRPO             | 46.46     | 56.66    | 35.72     | 50.00    | 21.66     | 23.33    | 47.39      | 60.00    | 37.81   | 47.50   |
> > | XRPO w/o AS      | 39.17     | 53.33    | 33.13     | 40.00    | 20.10     | 26.67    | 45.00      | 60.00    | 34.35   | 45.00   |
> > | XRPO w/o ICL     | 44.58     | 53.33    | 35.94     | 43.33    | 20.94     | 20.00    | 46.04      | 56.67    | 36.87   | 43.33   |
> > | XRPO w/o AS+ICL  | 40.72     | 46.66    | 31.45     | 40.00    | 19.06     | 20.00    | 46.04      | 60.00    | 34.32   | 41.67   |
> > | Dynamic Sampling | 40.31     | 43.33    | 32.50     | 33.33    | 20.42     | 23.33    | 39.90      | 43.33    | 33.28   | 35.83   |
> > | TreePO Sampling  | 38.33     | 50.00    | 26.04     | 33.33    | 17.29     | 20.00    | 37.40      | 46.67    | 29.77   | 37.5    |

---

> > > ### Author Response · Authors · 2025-11-22
> > > **Response to Reviewer zmm5 (Part 3)**
> > >
> > > ***Q7: The ICL seeding corpus evolves during training, but its scalability and retrieval efficiency are not adequately analyzed.***
> > >
> > > We thank the reviewer for raising this important point. We clarify the design of the ICL corpus and its implications for scalability.
> > > - ***Scalability and retrieval efficiency.*** To keep the ICL seeding mechanism efficient, we precompute and store nearest-neighbor relationships between prompts before training. During RL training, retrieval is then implemented as a simple lookup into this precomputed table, rather than performing online similarity search at every step, which makes the overhead negligible compared to the backbone forward passes. As shown in Table R5, all ICL-related operations correspond to only ~1% overhead compared to vanilla GRPO.
> > > - ***Initialization and growth.*** We start from generations of the baseline model to seed the initial corpus and then augment it during training with high-quality successful rollouts.
> > > - ***Cold-start perspective.*** Our reported results assume a cold-start ICL corpus, i.e., we do not pre-populate it with stronger external models. Thus, the performance we report is a lower bound on what could be achieved if one were to initialize the corpus using stronger SFT or teacher models.
> > >
> > > ***Q8: Figure 4(a): Why does XRPO produce shorter response lengths than GRPO? The paper introduces no explicit mechanism that reduces generation length.***
> > >
> > > While XRPO does not include an explicit length penalty, the shorter response length is an emergent property of its improved exploration under context constraints. GRPO suffers from inherent "length bias", often generating long and repetitive reasoning chains. On complex problems, these runaway chains hit the model's maximum context limit, resulting in truncation and failure (zero reward). GRPO thus struggles to learn from these long and failed trajectories. XRPO's Rollout Allocator and In-Context Seeding actively guide the model toward valid solution paths for hard prompts. Since a correct solution must be complete within the context window, XRPO effectively filters for trajectories that are concise enough to succeed. This interpretation is further supported by the empirical results shown in Table R8 that ICL reduces average generation length by 34.7% for Qwen3 and 6.17% for Qwen2.5.
> > >
> > >
> > > ***Q9: How does the computational overhead of dynamic rollout allocation compare to static allocation?***
> > >
> > > Our new results (Table R5) indicate that the per-step latency ratio between XRPO and baseline GRPO is about 1.047, mere 4.7% overhead.  We use a batch size of 64, 256 rollouts per prompt, and two dynamic rounds to better simulate common training settings. The ICL corpus is loaded once at the start of training (4.22 seconds), and constructing ICL prompts is extremely fast (<0.001 seconds per prompt) due to simple hash-map lookups over precomputed neighbors and responses.
> > >
> > > Furthermore, we provide the theoretical analysis on XRPO’s latency compared to the baseline in Appendix D.
> > >
> > > Overall, XRPO not only substantially reduces the number of convergence steps but also maintains a similar per-step latency.
> > >
> > > ----
> > >
> > > Table R5: XRPO Latency analysis
> > >
> > > | Method | Dynamic Planning | Actor Rollout | Construct ICL Prompts | Update ICL Corpus | Advantage Shaping |   Total  |
> > > |:------:|:----------------:|:-------------:|:---------------------:|:-----------------:|:-----------------:|:--------:|
> > > |  XRPO  |      79.497      |    1448.552   |         6.699         |       11.816      |       0.329       | 1546.893 |
> > > |  GRPO  |         -        |    1477.861   |           -           |         -         |         -         | 1477.861 |
> > >
> > > Table R8: ICL reduces average generation length
> > > | Response length | Qwen3 | Qwen2.5 |
> > > |-----------------|-------|---------|
> > > | Without ICL     | 881.7 | 12906.1 |
> > > | With ICL        | 827.2 | 8427.9  |
> > > | Reduction       | 6.17% | 34.7%   |

---

> > > > ### Author Response · Authors · 2025-11-22
> > > > **Response to Reviewer zmm5 (Part 4)**
> > > >
> > > > Table R3: Comparison of GSPO, Dynamic Sampling and TreePO Sampling with XRPO across benchmarks for Qwen3-1.7B (Reasoning).
> > > > |                  |   AIME 	   |    2024   |   AIME    |    2025   |  HMMT  |    2025   |   BRUMO   |    2025   | CodeForces |           |  Average  |
> > > > |------------------|:---------:|:---------:|:---------:|:---------:|:------:|:---------:|:---------:|:---------:|:----------:|:---------:|:---------:|
> > > > |                  |   pass@1  |  const@32 |   pass@1  |  const@32 | pass@1 |  const@32 |   pass@1  |  const@32 |   pass@1   |   pass@1  |  const@32 |
> > > > |       GSPO       |   42.39   | 60.00 |   33.23   |   46.67   |  21.25 |   20.00   |   45.20   |   56.67   |    13.72   |   31.16   |   45.84   |
> > > > | Dynamic Sampling |   40.31   |   43.33   |   32.50   |   33.33   |  20.42 |   23.33   |   39.90   |   43.33   |    9.48    |   28.52   |   35.83   |
> > > > |  TreePO Sampling |   38.33   |   50.00   |   26.04   |   33.33   |  17.29 |   20.00   |   37.40   |   46.67   |    9.51    |   25.71   |   37.50   |
> > > > |       XRPO       | 46.46 |   56.66   | 35.72 | 50.00 |  22.29 | 26.67 | 47.39 | 60.00 |  13.80 | 33.13 | 48.33 |
> > > >
> > > >
> > > > Table R4: Comparison of GRPO, Dynamic Sampling and TreePO Sampling  with XRPO across benchmarks for Qwen2.5-7B
> > > > |                     |        | AIME 2024 |          |        | AIME 2025 |          |        | BRUMO 2025 |          |        | Average |          |
> > > > |---------------------|--------|-----------|----------|:------:|:---------:|:--------:|:------:|:----------:|:--------:|:------:|:-------:|:--------:|
> > > > |                     | pass@1 | pass@4    | const@32 | pass@1 | pass@4    | const@32 | pass@1 | pass@4     | const@32 | pass@1 | pass@4  | const@32 |
> > > > | GRPO                | 10.31  | 18.10     | 13.33    | 6.73   | 15.31     | 10       | 15.83  | 26.30      | 26.67    |  10.96 |  19.90  |   16.67  |
> > > > | Dynamic Sampling    | 10.52  | 20.22     | 16.67    | 6.67   | 15.57     | 10       | 17.92  | 28.72      | 26.67    |  11.70 |  21.50  |   17.78  |
> > > > | TreePO Sampling     | 8.96   | 17.20     | 16.67    | 5.10   | 13.72     | 6.67     | 19.17  | 29.55      | 26.67    |  11.08 |  20.16  |   16.67  |
> > > > | XRPO                | 11.25  | 21.17     | 16.67    | 7.71   | 16.80     | 13.33    | 17.19  | 29.25      | 30.00    |  12.05 |  22.41  |   20.00  |
> > > > | Rel. Gain wrt GRPO. | 9.09%  | 16.96%    | 24.98%   | 14.55% | 9.73%     | 33.30%   | 8.55%  | 11.22%     | 12.53%   |        |         |          |

---

### Author Response · Authors · 2025-11-23
**Updated Manuscript and Highlighted Revisions**

Thank you reviewers for your thoughtful feedback. We have updated the PDF to include the new results and have highlighted all changes in blue for easier reference. We appreciate your time and consideration, and we hope the revisions address the points raised in the reviews.  We are happy to discuss further and help address any other concern.

---

### Author Response · Authors · 2025-11-25
**Request for Reviewer Feedback on Responses**

Dear reviewers,

Thank you for the time and attention you have given to evaluating our submission. We have now posted responses addressing all points raised in the reviews, and we would greatly appreciate it if you could take a moment to look them over and share any followup thoughts or updates to your assessment. Please let us know if any part of our responses would benefit from further clarification.

---

### Comment · Area_Chair_T3Ui · 2025-11-26
**Reminder of the Discussion**

Hi all reviewers,

Remember to attend the author-reviewer discussion, and your feedback is valuable. Note the ddl is Dec. 3rd.

Best,
AC

---

### Author Response · Authors · 2025-12-02
**Overview of the Rebuttal on XRPO**

Dear AC,

We sincerely thank all reviewers for their thoughtful assessments and valuable suggestions. Their insights significantly improved the clarity, rigor, and overall presentation of our work. We are also grateful to the area chair for guidance and timely support throughout the review and rebuttal process.

We are grateful that the reviewers recognized several positive aspects of our method XRPO, including: “systematic approach to tackle exploration and exploitation issue is RLVR (TrCK, SgHu, Krpv), “strong empirical performance” (zmm5, Krpv, TrCK),“insightful use of ICL to revive prompts” (Krpv, TrCK), “strong ablations and sensitivity studies” (Krpv), and “study and address a critical problem” (All reviewers).
For the comments and concerns raised in the reviews, we have prepared individual, ***point-by-point responses for each reviewer to address all the questions***. We have also revised the manuscript accordingly, and we summarize the key changes below. To facilitate cross-referencing, all modifications are highlighted in blue in the updated version.

Overall, the revised paper includes **3 additional pages** for new discussion and **7 new tables**, reflecting the substantial clarifications, analyses, and improvements inspired by the reviewers’ feedback.

## Paper Revision Summary:
***1. Additional Evaluation and Analysis.***

*  ${\color{blue} \texttt{Table 1}}$: Added Dynamic Sampling and TreePO Sampling baselines on Qwen3-1.7B (Reasoning) and Qwen2.5-7B-Instruct (suggested by Krpv and SgHu)
* ${\color{blue} \texttt{Table 2}}$: Added experiments on the new Llama-3.2-3B backbone (suggested by Krpv and SgHu).
* ${\color{blue} \texttt{Table 3 (b)}}$: Added sensitivity analysis for exploration strength $\lambda$ and confidence $\alpha$ (suggested by TrCK, SgHu, and zmm5).
* ${\color{blue} \texttt{Appendix B / Table 4}}$: Expanded ablation of XRPO components and comparisons with baselines (suggested by Krpv and zmm5).
* ${\color{blue} \texttt{Table 5}}$: Added Dynamic Sampling and TreePO Sampling baselines for Qwen2.5-7B-Instruct (suggested by Krpv and SgHu).
* ${\color{blue} \texttt{Section 5.2}}$: Added discussion of XRPO training overhead (suggested by all reviewers).
* ${\color{blue} \texttt{Appendix D}}$: Added mathematical justification for XRPO’s negligible training overhead (suggested by all reviewers).
* ${\color{blue} \texttt{Appendix F.1 / Table 6}}$: Added analysis of how novelty-guided advantage sharpening affects response length (suggested by Krpv, TrCK, and SgHu).
* ${\color{blue} \texttt{Appendix F.2/ Table 7}}$: Added analysis of how ICL affects response length (suggested by TrCK and zmm5).

***2. Writing Improvements.***
* ${\color{blue} \texttt{Section 5.1}}$: Clarified training dataset composition and added details on new baselines and models.
* ${\color{blue} \texttt{Section 5.2}}$: Added discussion of Llama-3.2-3B results and training overhead.
* ${\color{blue} \texttt{Section 5.3}}$: Expanded discussion of hyperparameter robustness.

We again thank all reviewers for their constructive feedback, which has helped us significantly strengthen the paper. We hope that our responses and the revised manuscript address all concerns, and we would be happy to provide any further clarification.

Thank you,

**Authors of XRPO**

---

### Meta-Review · Area_Chair_BC9o · 2025-12-30

**Summary:**

XRPO extends GRPO with three mechanisms (hierarchical rollout planning, ICL seeding for zero-reward prompts, and "novelty-guided" advantage sharpening) aiming to better balance exploration/exploitation and improve RLVR training. Reviewers agree the problem is important and the paper reports broad empirical gains, but two reviewers raised major concerns: (i) the contribution feels like a packaging of existing ideas with overstated “unified framework”/novelty claims, and (ii) the approach adds substantial complexity and hyperparameter burden that weakens practical appeal. Additionally, Reviewer SgHu’s post-rebuttal comment indicates the added experiments/clarifications still do not convincingly establish that each component adds value beyond strong baselines, nor that ICL seeding constitutes a genuinely new exploration mechanism (vs. replay/augmentation), and reported gains vs dynamic sampling remain small on some key metrics.

I have carefully read the reviews and the discussions, as well as the paper itself. Although I commend the authors efforts in the rebuttal with additional explanations, tables and analyses, my own overall assesment of the paper agrees with the concerns raised by Reviewers zmm5 and SgHu, as summarized above. I thus recommend rejection.

**Reviewer Concerns:**

Addressed:

* stronger baselines and new models

* robustness clarity

* training overhead

Outstanding:

* Novelty / unified framework overclaiming
* Component-level value beyond strong baselines
* ICL mechanism interpretation
* heuristic nature of blending together different components

**Reviewer Scores:**

zmm: 4 --> 4 (no participation)
Krpv: 6--->6 (no participation)
SgHu: 2---> 2 (participated partially in discussion)
TrCK: 6--->6 (no participation)

---

### Decision · Program_Chairs · 2026-01-26

Reject